# Stacking transfer of wafer-scale graphene-based van der Waals superlattices

Guowen Yuan [1,3], Weilin Liu[1,3], Xianlei Huang[1], Zihao Wan[1], Chao Wang[1], Bing Yao[1], Wenjie Sun[2], Hang Zheng[1], Kehan Yang[1], Zhenjia Zhou[1], Yuefeng Nie [2], Jie Xu [1] & Libo Gao [1] ✉

High-quality graphene-based van der Waals superlattices are crucial for investigating physical properties and developing functional devices. However, achieving homogeneous wafer-scale graphene-based superlattices with controlled twist angles is challenging. Here, we present a flat-to-flat transfer method for fabricating wafer-scale graphene and graphene-based superlattices. The aqueous solution between graphene and substrate is removed by a two-step spinning-assisted dehydration procedure with the optimal wetting angle. Proton-assisted treatment is further used to clean graphene surfaces and interfaces, which also decouples graphene and neutralizes the doping levels. Twist angles between different layers are accurately controlled by adjusting the macroscopic stacking angle through their wafer flats. Transferred films exhibit minimal defects, homogeneous morphology, and uniform electrical properties over wafer scale. Even at room temperature, robust quantum Hall effects are observed in graphene films with centimetre-scale linewidth. Our stacking transfer method can facilitate the fabrication of graphene-based van der Waals superlattices and accelerate functional device applications.

Two-dimensional (2D) van der Waals superlattices (vdWS), which are vertically stacked by individual 2D layered materials, offer a platform for exploring physical properties and functionality[1,2]. Based on the vdWS assembled from 2D conductors, semiconductors, ferromagnets and superconductors, a series of functional devices have been constructed, such as vertical field effect tunnelling transistors[3], superconducting Josephson junctions[4], spintronic memories[5] and photodetectors[6]. Nowadays, there are mainly two strategies to build 2D vdWS with wafer scale, including the bottom-up stacking growth[7,8] and the top-down transfer[9,10]. Therein, chemical vapour deposition (CVD) growth of various large-sized transition metal dichalcogenides (TMDCs) heterostructures has been demonstrated[7,8]. Moreover, benefiting from the weak coupling between the wafer-scale TMDCs with the growth substrate, one layer-by-layer transfer method of namely

programmed vacuum stack has also been developed for building the TMDCs-based vdWS[10].

Graphene, as the first discovered and highly conductive 2D material with remarkable mechanical and electrical properties, is likely to be the most compatible component of future vdWS and their functional devices[11]. Furthermore, graphene-based vdWS have also revealed numerous fresh physical properties. Such superlattice minibands of Hofstadter's butterfly[12,13], unconventional superconductivity[14], correlated insulator behaviour[15], ferromagnetism[16], quantized anomalous Hall effect[17] etc., are stimulated in the stacked graphene/hBN or double-layer graphene by precisely controlling their twist angles. Graphene-based vdWS offer possibilities for creating a generation of ultrathin transparent functional devices[3,18]. However, there are a few effective methods to stack growth double-layer graphene with

[1]National Laboratory of Solid State Microstructures, Jiangsu Key Laboratory for Nanotechnology, School of Physics, Collaborative Innovation Center of Advanced Microstructures, Nanjing University, Nanjing, China. [2]National Laboratory of Solid State Microstructures, Jiangsu Key Laboratory of Artificial Functional Materials, College of Engineering and Applied Sciences and Collaborative Innovation Center of Advanced Microstructures, Nanjing University, Nanjing, China. [3]These authors contributed equally: Guowen Yuan, Weilin Liu. ✉e-mail: lbgao@nju.edu.cn

controllable twist angles[19,20], but the stacking growth strategy is usually hindered by the harsh growth conditions. Wafer-scale graphene films are typically grown via CVD method at a relatively high temperature (normally >800 °C)[21], which is much higher than most of other 2D materials. Therefore, graphene is commonly used as the underlying layer while growing in the stack manner[22,23], which makes it difficult to integrate graphene into the programmed order of vdWS. In contrast, the layer-by-layer transfer shows more flexibility to build diverse graphene-based vdWS. Actually, most graphene-based superlattices with specific properties are fabricated through the modified dry transfer technique[24], where the graphene or other 2D materials are limited to dozens of microns. Until now, it is relatively easy to realize the transfer of wafer-scale monolayer graphene[25–34], and there are also a few approaches for manipulating the twist angles between the adjacent graphene layers[35–37]. However, stacking homogeneous graphene-based vdWS at wafer size remains a challenge, and the homogeneous vdWS with controlled twisted angles is more difficult.

During traditional operations, the macroscopical transfer defects are the main trouble for the homogenous feature and the subsequent stacking transfer[25,38,39]. Wrinkles or some folds, which are formed during CVD growth at high temperature, can be eliminated by the proton-assisted or low-temperature methods[21,40]. However, most folds, cracks and tears are induced by the inappropriate transfer operation of pasting graphene on the target substrate or other 2D materials[25,27,38], where the trapped water or residual nanoparticles (NPs) should be the main obstacles[30–32,41]. Once graphene is firmly attached to substrate, it is difficult to reduce the trapped water, NPs and consequent folds through the process like nitrogen gas blow[31,32,42]. The adsorbed water also causes the heavily doping state of graphene and degenerates their electrical performance[43]. Thus, the transfer defects and the residues on the bottom 2D materials should be completely eliminated if stacking transfer a homogeneous vdWS. Furthermore, the hydrophilic substrates show more helpful for transferring defect-free graphene films than the hydrophobic ones[27,30], the substrates with flatter surfaces are also better than the rough ones[33,44], and the residual NPs trapped in the interlayers of stacked vdWS are considered to be avoided through the layer-by-layer hydrogen annealing[45]. However, most 2D materials are approximately hydrophobic, and the wetting angle of DI water on intrinsic graphene ($\theta_1$) is more than 80 degree[46], hence the high-quality stacking transfer of wafer-scale graphene-based vdWS is still plagued by the trapped air, water or NPs in the interlayers.

In this work, we develop a flat-to-flat transfer method, i.e. transferring flat graphene films onto the flat substrate, which consists of the spinning-assisted process and the proton-assisted treatment (PAT) process which is based on the permeable protons and recombined hydrogen, to exhaust the trapped water and decouple the transferred graphene. The wafer-scale monolayer graphene and graphene-based vdWS with ultra-flat surface and scarcely transfer defects are obtained through adjusting the appropriate wetting angles in different aqueous solutions. The twist angles of double-layer and triple-layer graphene can be precisely adjusted by the macroscopically stacking angles according to the wafer flats of graphene. The transferred graphene and graphene-based vdWS exhibit excellent morphological, structural, optical, and electrical homogeneity over the wafer size. Robust quantum Hall effect (QHE) arises in the transferred graphene films, and even appears at room temperature (RT, normally 298 K) with linewidth of centimetres.

## Results

### Flat-to-flat transfer process

Figure 1 illustrates the schematic of the flat-to-flat method to transfer ultra-fat graphene film to flat SiO$_2$/Si under wet condition. Because the wetting angle of DI water on SiO$_2$/Si ($\theta_2$) is usually less than 60 degree[30], the approximative hydrophilia makes DI water can be used as the transfer solution here (Fig. 1a). Inspired by the spin-coating procedure in semiconductor technology, a two-step spinning-assisted transfer process is developed to equably eject the trapped water layers based on the centrifugal pulls. The slow spinning-assisted dehydration is firstly performed, where the trapped thick H$_2$O can be uniformly reduced to be a thinner layer. If the substrate is hydrophobic, like graphene or other 2D material films, the DI water should be replaced by other solution, like mixing IPA in aqueous solution, and then graphene becomes hydrophilic (Fig. 1b, wetting angle $\theta_3$). Atomic force microscope (AFM) measurements show that the thickness of the DI water layer trapped between graphene and SiO$_2$/Si is ~40 nm and the roughness of PMMA/graphene/DI water/SiO$_2$/Si is ±25 nm (supplementary Fig. 1). Then, a fast spinning-assisted dehydration is followed to further remove the trapped solution. The graphene films are homogenously pasted on the substrate, and the trapped water bubbles and extruded folds are both avoided ultimately.

After the two-step spinning-assisted dehydration and baking process, the wafer-scale graphene/SiO$_2$/Si can be integrally obtained after removing PMMA. Normally, the surface of graphene film at this stage often remains a large number of residual NPs, like undissolved PMMA, adsorbed ambient acetone or water. Particularly, there should be residual water molecules (H$_2$O) trapped in the interface between graphene and substrate, which are difficult to be eliminated through the thermal annealing (Fig. 1c). All the adsorbed NPs or trapped H$_2$O

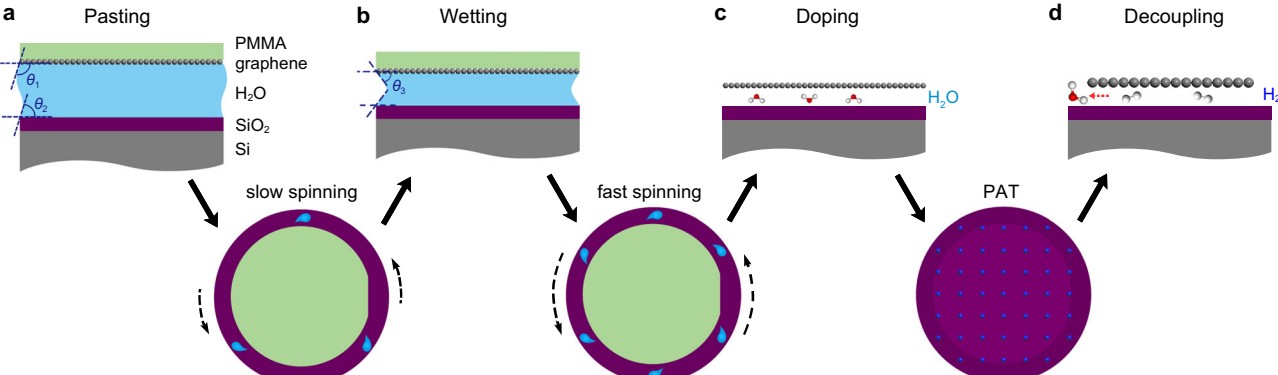

**Fig. 1 | Schematic of the flat-to-flat transfer process.** The transfer process includes a two-step spinning-assisted dehydration process and a proton-assisted treatment (PAT) decoupling process. **a** The different stages represent pasting polymethyl methacrylate (PMMA)/graphene onto SiO$_2$/Si wafer. **b** The thinning solution layer between PMMA/graphene and SiO$_2$/Si after slow spinning process. **c** Doping graphene films by the residual water. **d** Neutralizing graphene films by the PAT decoupling process. $\theta_1$ is the wetting angle of DI water on intrinsic graphene, graphene is hydrophobic. $\theta_2$ is the wetting angle of DI water on SiO$_2$/Si, which is usually less than 60 degree. $\theta_3$ is the wetting angle of revised solution, such as IPA/water solution, on intrinsic graphene, here graphene becomes hydrophilic.

molecules cause the heavy doping state of graphene, reduce the mean free path of high-mobility carriers, and finally compromise the electrical performances. As supplementary to the flat-to-flat transfer, we utilize the PAT process to clean and decouple graphene homogenously (Fig. 1d), the permeated protons and re-bonded $H_2$ should play an important role in decoupling graphene. After the comprehensive flat-to-flat transfer process, the wafer-scale graphene films on a flat substrate with ultra-clean and ultra-flat surface can be obtained.

**Transferring wafer-scale monolayer graphene**
Figure 2a shows a typical photo of 4-inch graphene on $SiO_2$/Si wafer transferred by the flat-to-flat method, where the film looks homogenous at the macroscopic scale and can be distinguished from the reserved blank region. The enlarged optical image is shown in Fig. 2b, there are no cracks, folds and tears in the film and graphene can be only distinguished from the tweezer scratched blank, indicating the transferred film presents excellent optical homogeneity. Figure 2c is the typical AFM height image across the scratches, showing that the thickness of the transferred monolayer graphene film is homogenous with a height of ~1.1 nm. More information about the wafer-scale graphene with scarcely any transferred defects and the ex situ AFM images for the as-grown and as-transferred graphene films are shown in supplementary Fig. 2. These results show that the morphology of the as-grown graphene films can be perfectly preserved without any damage during this flat-to-flat transfer process. Figure 2d compares the surface cleanness of as-transferred graphene films after different treatments, including the vacuum annealing and the PAT processes. The statistical chart of residual NPs over $15 \times 15$ μm² indicates that the as-transferred graphene contains large amounts of residues with an average height of ~4 nm, the vacuum annealing process will remove most NPs with height of ~3 nm. However, the PAT process can completely clean the surface of graphene films, and nearly no NPs are found. Their corresponding AFM images are compared in supplementary Fig. 3.

Subsequently, the statistical distribution of Raman intensity ratio of 2D to G peaks ($I_{2D}/I_G$) over $100 \times 100$ μm² of transferred graphene after different treatments are shown in Fig. 2e, inset is the corresponding Raman mapping of the graphene after PAT. It shows that the $I_{2D}/I_G$ of the as-transferred graphene is concentrated at the values of $1.6 \pm 0.3$, demonstrating its monolayer nature. The $I_{2D}/I_G$ decreased to $1 \pm 0.2$ after vacuum annealing, mostly caused by the annealing introduced substrate doping[47]. In contrast, the distribution of $I_{2D}/I_G$ after PAT is more uniform and exhibits minimal variation. Typical Raman mapping for $I_{2D}/I_G$ of the as-transferred graphene and graphene after annealing are shown in supplementary Fig. 4a, b, and Raman spectra, Raman mapping for the full width at half maximum (FWHM) of G peak and the $I_D/I_G$ of the graphene after PAT are plotted in supplementary Fig. 4c–e. In addition, the in situ variable temperature Raman measurements from RT to 200 °C are also used to compare the coupling interaction between graphene and the substrate after different treatments[21]. Figure 2f plots the statistical distribution of the extracted shift values of 2D peak ($\Delta\omega_{2D}$), and it shows that the as-transferred graphene presents the least value of $\Delta\omega_{2D}$, indicating the weakest coupling between graphene and the substrate. After vacuum annealing, the $\Delta\omega_{2D}$ become the largest, meaning that the interaction is significantly enhanced. In contrast, $\Delta\omega_{2D}$ after PAT process present much smaller, indicating the PAT graphene remains the similar coupling state with the as-transferred film, and PAT is able to decouple graphene from the substrate and weaken their coupling. The detailed spectra are plotted in supplementary Fig. 4f–h.

We continue to measure the doping levels of transferred graphene films after different treatments via the back-gate field effect transistor (FET) devices, which are batch fabricated by the simple etching through a shadow mask and present the same Hall bar configurations. Figure 2g plots the transport properties of the transferred

graphene films after different treatments as a function of gate voltage ($V_{bg}$), and the inset shows the typical optical image of the Hall bars with 200 μm linewidth. The as-transferred graphene films show the heaviest $p$-doping level with the carrier density of $4.4 \times 10^{12}$ cm⁻², where the Dirac point is located at $V_{bg}$ of 58 V. This should reasonably be attributed to the large amounts of residual NPs and the adsorbed $H_2O$ molecules. After the vacuum annealing, the Dirac point of graphene film shifts to 16 V, i.e. the $p$-doping level is significantly reduced. In contrast, the graphene films after PAT process exhibits the weakest doping level with the carrier density of $2.9 \times 10^{11}$ cm⁻² and Dirac point at 3.8 V, indicating that PAT process can truly decouple graphene and neutralize the doping level. The statistical distribution of Dirac points and the corresponding carrier density of the transferred graphene films are compared in supplementary Fig. 5a, and the PAT graphene films have the lowest doping level and remain the best intrinsic state. The FET mobilities of the Hall devices in Fig. 2g are extracted in supplementary Fig. 5b. The as-transferred graphene shows the lowest hole mobility of ~4100 cm² V⁻¹ s⁻¹, and the hole mobility increases to ~6950 cm² V⁻¹ s⁻¹ after vacuum annealing. In contrast, the PAT graphene presents the best state, where the hole and electron mobilities reach ~13,000 cm² V⁻¹ s⁻¹ and ~10,600 cm² V⁻¹ s⁻¹ at 1.5 K, and they also reach ~7450 cm² V⁻¹ s⁻¹ and ~7250 cm² V⁻¹ s⁻¹ at RT, respectively.

The realizations of QHE usually represent the high electrical quality of transferred graphene films. The longitudinal resistances ($R_{xx}$) and Hall conductivity ($\sigma_{xy}$) for the same Hall devices in Fig. 2g measured under the constant perpendicular magnetic field ($B_\perp$) of 6.5 T and the temperature of 1.5 K are plotted in supplementary Fig. 5c. The QHE for the PAT graphene film is robust, with more emergent integer plateaus and highly accurate plateau values. Moreover, the Hall devices with different linewidths fabricated by the PAT graphene films present nearly the same thresholds to realize QHE in supplementary Fig. 5d, indicating that the emergence of QHE is almost independent of the linewidth for PAT graphene. Just because of the high homogeneity, we can fabricate the centimetre-sized Hall devices, which reaches the size limit in our measuring equipment. The typical photo is shown in Fig. 2h, and this should be the largest measured graphene Hall bar to our best knowledge. Figure 2i plots the $R_{xx}$ and $\sigma_{xy}$ at different temperatures, where the first $\sigma_{xy}$ plateaus at $\pm 2e^2/h$ easily appears even at RT and the fifth integer plateaus is still obvious under $B_\perp$ of 7 T and at 1.5 K. The robust QHE plateaus emerged under relatively mild measurement conditions demonstrate the high homogeneity and high quality of the flat-to-flat transferred films on a wafer scale. Moreover, the hole mobilities of this centimetre-sized graphene film are still up to ~12,000 cm² V⁻¹ s⁻¹ at 1.5 K and ~9300 cm² V⁻¹ s⁻¹ at RT, as shown in supplementary Fig. 5e, f.

**Stacking transfer of wafer-scale few-layer graphene**
The flat-to-flat method is easily generalized to the stacking transfer of wafer-scale few-layer graphene-based vdWS. Figure 3a displays a typical photo of triple-layer graphene films obtained by sequentially stacking 4-inch graphene films onto a 6-inch $SiO_2$/Si wafer, and the stacked films present homogenous on the macroscopic scale and there are no obvious transfer defects. Different from the transfer of monolayer graphene, the DI water is not suitable for the stacking transfer of graphene-based vdWS. The wetting angle of the used solution on graphene is crucial, corresponding to the step in Fig. 1b. If the wettability for graphene ($\theta_1$) and the substrate ($\theta_2$) are both hydrophobic, the folds are frequently formed in the stacked graphene films, and the typical optical image is shown in the top inset of Fig. 3b. Considering this, we start to tune the wetting angles through mixing a certain concentration of IPA or acetic acid into DI water (see Methods), and the wetting angles with different IPA concentrations in DI water are plotted in supplementary Fig. 6a. We summarize the dependence between the fold density of stacked graphene and the wetting angles in Fig. 3b, and the optimum wetting angles for the stacked graphene-based vdWS are

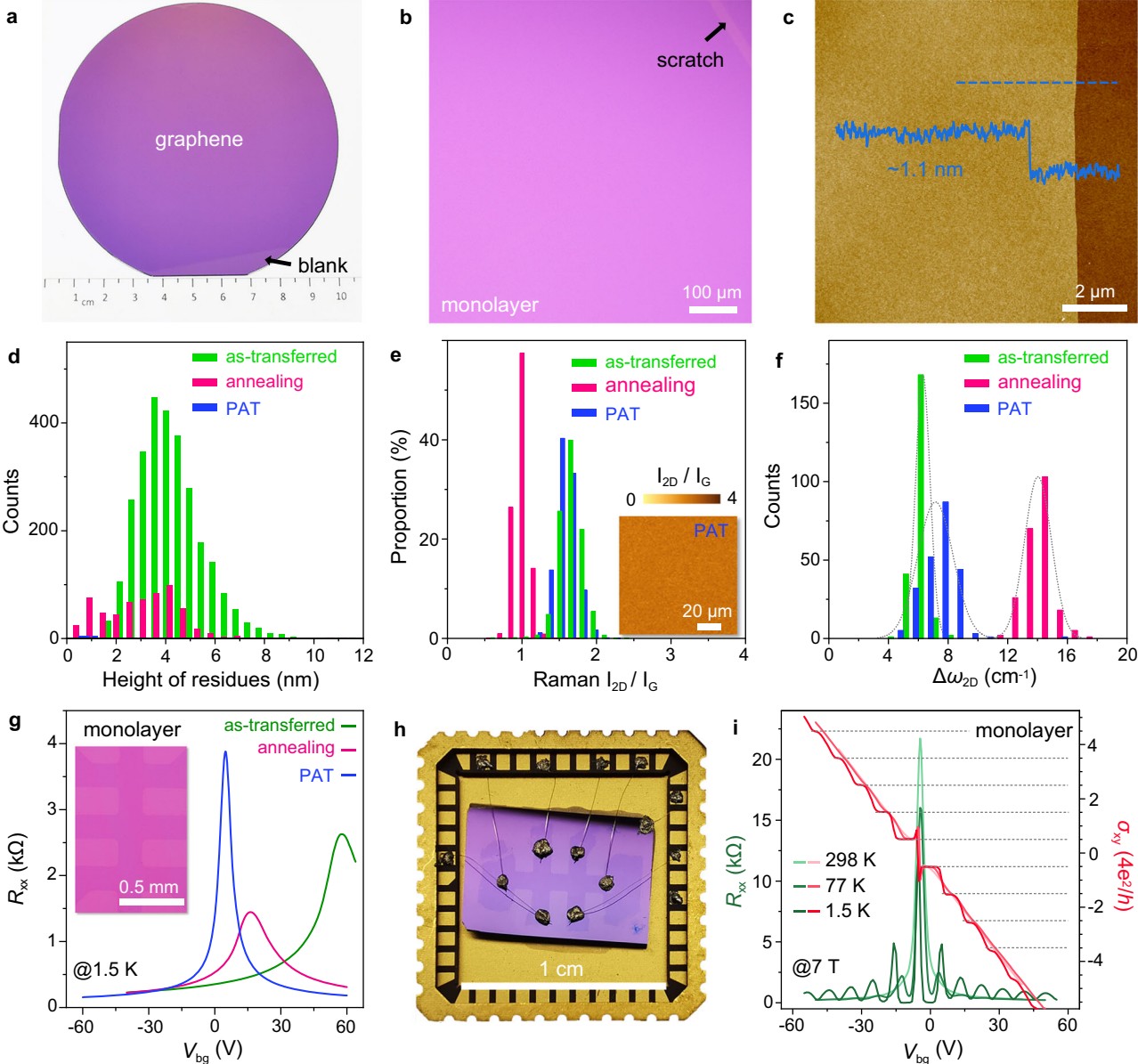

**Fig. 2 | Flat-to-flat transfer of wafer-scale monolayer graphene films. a** Typical photo of 4-inch graphene transferred onto $SiO_2$/Si wafer, the film looks homogenous at the macroscopic scale and the graphene can be distinguished from the reserved blank region. **b** Typical optical image of graphene on $SiO_2$/Si, there are no cracks, folds, tears in the film and graphene can be only distinguished from the scratch. **c** Typical AFM height image of graphene and the inset height profile is taken along the dashed line, the thickness is homogenous with ~1.1 nm. **d** Statistical chart of the residual nanoparticles (NPs) on the as-transferred graphene, after vacuum annealing and after PAT, the annealing process will remove some NPs and the PAT can make graphene films much cleaner. **e** Statistical distribution of Raman intensity ratio of 2D to G peak ($I_{2D}/I_G$) in $100 \times 100$ $\mu m^2$ of the as-transferred graphene, after vacuum annealing and after PAT, inset is the corresponding Raman mapping of the graphene after PAT. The distribution of $I_{2D}/I_G$ is homogeneous after PAT with the value barely unchanged, while annealing will cause the decrease of the values. **f** Statistical distribution of the extracted shift values of 2D peak ($\Delta\omega_{2D}$)

measured from room temperature (RT) to 200 °C for the as-transferred, after vacuum annealing and after PAT graphene, the grey lines are the fit curves of the normal distributions. The PAT results in the similar coupling state to the as-transferred film. **g** Longitudinal resistances ($R_{xx}$) of the as-transferred, after vacuum annealing and after PAT graphene films as a function of gate voltage ($V_{bg}$), all the samples are fabricated into the large-sized Hall bars and the inset is the typical optical image. The PAT process can well decouple graphene and neutralize the doping level. **h** Photograph of the centimetre-sized Hall bar fabricated by transferred graphene on $SiO_2$/Si after PAT, which is fixed on a chip carrier. **i** Measured quantum Hall effect (QHE) for the centimetre-sized graphene Hall bars in **h**. $R_{xx}$ and Hall conductivity ($\sigma_{xy}$) as a function of $V_{bg}$ at different temperatures under perpendicular magnetic field $B_\perp = 7$ T, the horizontal dashed lines are the guide lines of the Hall plateaus. Robust Hall plateaus can be easily observed even at RT, indicating the homogenous electrical properties over the wafer scale.

from 40° to 60°. Thereafter, combining with the two-step spinning-assisted transfer processes and the wetting angle of ~53° (bottom inset in Fig. 3b), the folds and cracks are successfully avoided and the stacked double-layer graphene films with wafer scale are fabricated. The detailed optical images of transferring double-layer graphene with different wetting angles are compared in supplementary Fig. 6b–d.

After spinning-assisted transfer, the stacked graphene film is also treated by PAT process. Typical AFM images of the stacking transferred double-layer graphene are shown in Fig. 3c and supplementary Fig. 7a, and their surfaces are flat and clean, and the thickness is homogenous with ~1.5 nm. Supplementary Fig. 7b also displays the stacking transferred triple-layer graphene with a homogenous

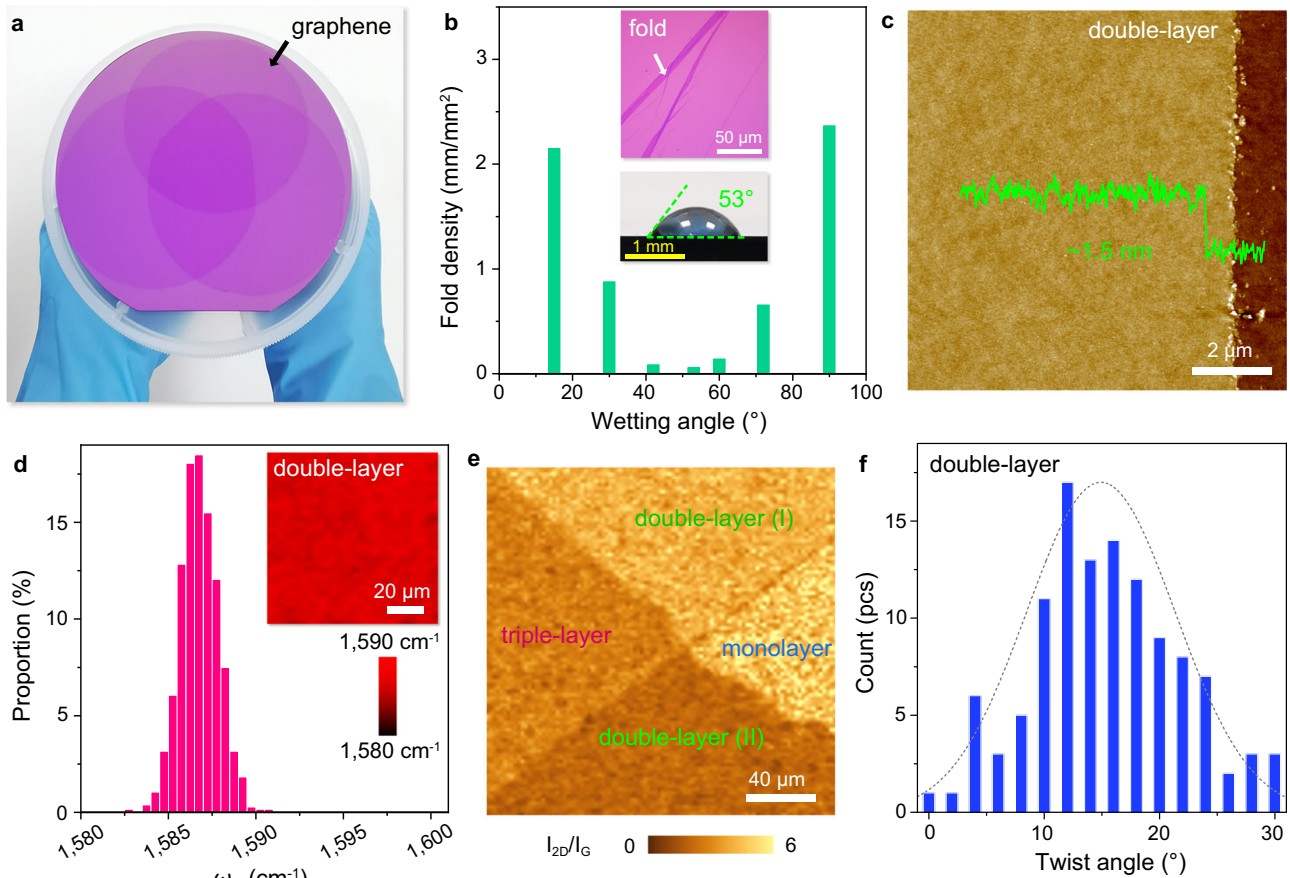

**Fig. 3 | Stacking transfer of wafer-scale double-layer and triple-layer graphene films. a** Photo of triple-layer graphene films obtained by sequential stacking 4-inch graphene to a 6-inch SiO₂/Si wafer, indicating the film is homogenous on the macroscopic scale and there are no obvious folds. **b** Dependence between fold density and the wetting angle, the top inset is the typical optical image of graphene folds and the bottom inset is the photograph for appropriate wetting angle of ~53° to avoid the formation of folds. **c** Typical AFM height images of double-layer graphene and the inset height profile is taken along the dashed line, the thickness is homogenous with ~1.5 nm. **d** Statistical distribution of the value of G peak ($\omega_G$) across 100 × 100 μm² of double-layer graphene, the inset is the corresponding Raman mapping. **e** Raman mapping of the $I_{2D}/I_G$ across the monolayer, double-layer and triple-layer transition regions with area of 200 × 200 μm². The distribution is homogenous in the same stacked regions of double-layer and triple-layer graphene, and is distinguished significantly in different stacked layers. **f** Statistical distribution of twist angles about the multiple double-layer graphene that obtained randomly and the dashed line is the fit curve, the twist angles follow the normal distribution.

thickness of ~2.1 nm. It is worth noting that the ex situ AFM images in supplementary Fig. 7c show that the PAT process can also clean the interface between graphene and the substrate, and the trapped polymer NPs or H₂O clusters could be reduced and excluded from the graphene edges during the PAT process. Furthermore, the residues trapped in the interlayer of double-layer graphene can also be reduced and the double-layer graphene is further flattened through the PAT process. Double-layer graphene films during the multi-cycled PAT processes are non-destructive, and the Ar plasma treatment seems ineffective in eliminating the intralayer residues. All the evolutional results are compared in supplementary Fig. 7d–h, indicating that the permeated protons should play an important role in the interface cleaning.

Figure 3d and supplementary Fig. 8a, b show the values of $\omega_G$ and $\omega_{2D}$ for double-layer graphene films, and their distributions are both within the range of ±2.5 cm⁻¹ in large area, indicating high homogeneity. The typical Raman mapping of $I_{2D}/I_G$, FWHM of G and 2D bands, $\omega_G$ and $\omega_{2D}$ across the monolayer, double-layer and triple-layer transition regions are shown in Fig. 3e and supplementary Fig. 8c–h, respectively. Raman characters are significantly distinguished in different stacked layers, and the difference between double-layer (I) and double-layer (II) should be derived from their different twist angles (supplementary Fig. 9a)[48]. We further count hundreds of double-layer

graphene films which are stacked randomly and find their twist angles approximately follow the normal distribution, as shown in Fig. 3f.

## Controlling the twist angles of few-layer graphene films

Furthermore, this stacking transfer can be also applied to control the twist angles between the double-layer graphene films at wafer scale. Benefiting from the approximately single crystalline graphene films grown on Cu(111)/sapphire (supplementary Fig. 9b, c), we can distinguish the orientation difference during the pasting operation (step I in Fig. 1). Through adjusting the macroscopic stacking angle ($\alpha$) between the upper and bottom graphene films, we can preliminarily obtain a double-layer graphene film with a controlled twist angle, illustrated in Fig. 4a. The detailed procedures for controlling twist angles are shown in supplementary Fig. 10. The macroscopic stacking angle nearly remain unchanged during the whole processes, resulting in the wafer-scale double-layer graphene films with the designed stacking angle of ~20°. However, it is worth noting that the macroscopic stacking angle is calibrated by a high-resolution camera with the angle error of ±1°, thus the controllably acquired minimum twist angle of double-layer graphene should also be 0° with the error range of ±1° (supplementary Fig. 11).

To further demonstrate the controllability and repeatability of the macroscopic stacking angle, we have also transferred a wafer-scale

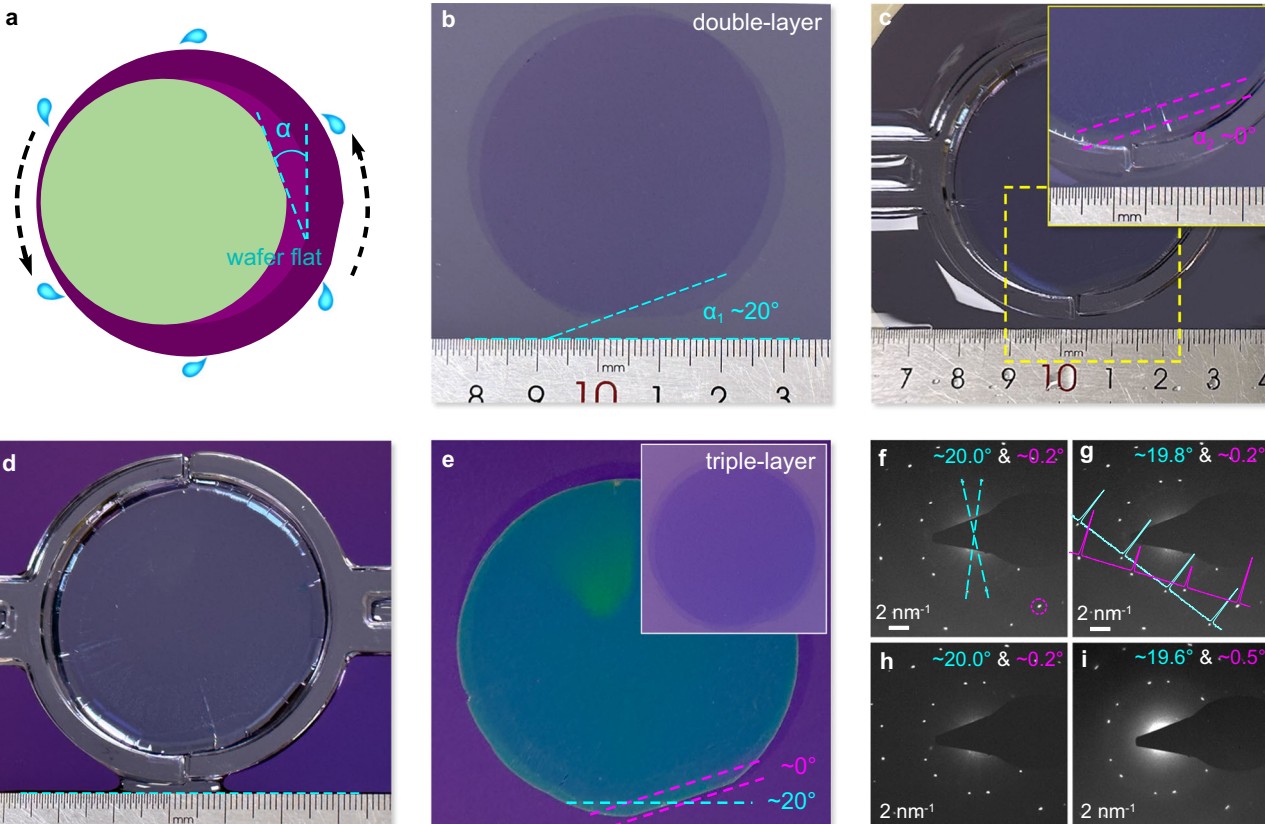

**Fig. 4 | Controlling the twist angle by adjusting the macroscopic stacking angle. a** Schematic of controlling the macroscopic stacking angle ($\alpha$) of double-layer graphene by rotating the orientations of graphene wafer flats. **b** Photo of the as-transferred double-layer graphene films with macroscopic stacking angle $\alpha_1$ of ~20°, immersing the film in the aqueous solution and adjusting one of the wafer-flat of graphene films parallel to the ruler. **c**, Adjusting the macroscopic stacking angle $\alpha_2$ at ~0° (i.e. ~20° to another graphene), inset is the zoom-in image of the yellow dashed box. **d** Exhausting the solution, descending the floating PMMA/graphene to finally achieve an initial alignment between the graphene layers. **e** Graphene films after spinning-assisted transfer process, and inset is the triple-layer graphene films after removing PMMA. **f–i**, SAED patterns of the triple-layer graphene collected at random locations. The twist angles measured by a dashed line and dashed circle in **f** are consistent with the macroscopic stacking angles. Insets in **g** are the relative intensities of the first-order and second-order SEAD spots, the purple and cyan lines represent the intensity profiles of the double-layer graphene with the twist angle of ~0° and the other monolayer graphene, respectively.

triple-layer graphene with designed twist angles. As shown in Fig. 4b, we first use a double-layer graphene with stacking angle $\alpha_1$ of ~20° as the bottom block and then immerse it into the aqueous solution. One wafer flat of bottom double-layer graphene is rotated in a specific orientation, which is usually parallel to a ruler. After that, we need to rotate the third floating graphene film and adjust another macroscopic stacking angle $\alpha_2$ to 0° before being fixed with support (Fig. 4c). This is similar to the operation in supplementary Fig. 11b-c. The solution is then slowly exhausted to descend the floating graphene, resulting in an initial alignment of the bottom block (Fig. 4d). Next, the fixers are removed and the solution between the graphene layers is also eliminated with the spinning-assisted transfer method. Figure 4e shows the stacked triple-layer graphene after the spinning process, and the macroscopic stacking angle remains unchanged at ~0° and ~20°.

The transferred double-layer and triple-layer graphene films are further analysed using selected area electron diffraction (SAED) patterns at random locations. Two pieces of 2-inch double-layer graphene films with designed stacking angles of ~6° and ~38°, as well as corresponding SAED patterns, are shown in supplementary Fig. 12a-e, and their twist angles and macroscopic stacking angles are basically consistent with each other. Figure 4f–i displays the SAED patterns of the obtained triple-layer graphene, the twist angles and macroscopic stacking angles are also consistent with each other. For twist angles of roughly 0°, the intensity ratios between the second-order and the first-order SAED spots are always larger than 2 (insets of Fig. 4g), indicating

the Bernal stacking order[20,49]. The variable temperature measurements confirm that the thermal annealing above 450 °C or PAT process will bring in the effective interlayer coupling and the stable twist angle (supplementary Fig. 12f). Moreover, we also transfer double-layer graphene films onto the flat Cu(111) substrate, then the millimetre-sized low-energy electron diffraction (LEED) is performed to confirm the uniformity of twist angles. There are only two sets of LEED patterns at various sites, and the twist angles are virtually identical, indicating that the twist angles are homogeneous across the wafer size (supplementary Fig. 12g). These results totally demonstrate that the twist angle can be homogenously controlled through the flat-to-flat transfer method.

## Transport characteristics of graphene-based vdWS

Subsequently, the double-layer and triple-layer graphene films with controlled twist angles are fabricated into Hall devices for evaluating the electrical properties. Figure 5a plots the $R_{xx}$ of the double-layer graphene with ~10° twist angle changes with varying $V_{bg}$ at 1.5 K, 77 K and RT, and the inset is the typical optical image of the Hall bar with 50 μm linewidth. This double-layer graphene film present slight $p$-type doping and the hole and electron mobilities reach ~7750 and ~7500 cm²V⁻¹s⁻¹ at 77 K, ~5950 and ~6250 cm² V⁻¹ s⁻¹ at RT, respectively (supplementary Fig. 13a). Figure 5b plots the QHE of this double-layer graphene, where the filling factors of Landau level at -3, -1, 1, 2 and 4 can be distinguished apparently at 1.5 K under $B_\perp$ of 7 T and these are

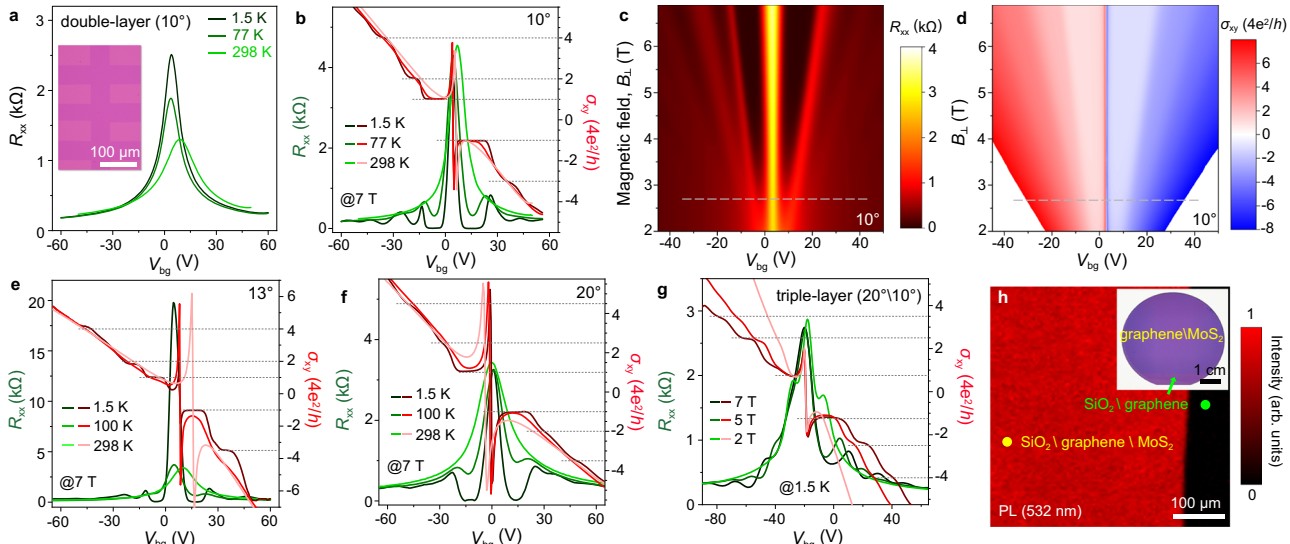

**Fig. 5 | Transport characteristics of stacking transferred double-layer graphene and other graphene-based van der Waals superlattices (vdWS). a** $R_{xx}$ of double-layer graphene as a function of $V_{bg}$ at 1.5 K, 77 K and RT, the twist angle is -10°. Inset is the optical image of the measured Hall bar with a linewidth of 50 μm. **b** $R_{xx}$ and $\sigma_{xy}$ of the double-layer graphene with -10° twist angle as a function of $V_{bg}$ under $B_\perp$ of 7 T at 1.5 K, 77 K and room temperature (RT), respectively. **c** 2D contour plot of $R_{xx}$ as a function of $V_{bg}$ and $B_\perp$ at 1.5 K about the same double-layer graphene in **a**, the Landau levels arise clearly when $B_\perp$ is larger than 2.7 T. **d** 2D contour plot of $\sigma_{xy}$ as a function of $V_{bg}$ and $B_\perp$ at 1.5 K, the robust QHE arises even when $B_\perp$ is lower to 2.7 T. **e** $R_{xx}$ and $\sigma_{xy}$ of double-layer graphene with -13° twist angle as a function of $V_{bg}$

under $B_\perp$ of 7 T at 1.5 K, 100 K and RT, respectively. **f** $R_{xx}$ and $\sigma_{xy}$ of double-layer graphene with twist angle of -20°, measured under $B_\perp$ of 7 T and at the temperature of 1.5 K, 100 K and RT, respectively. **g** $R_{xx}$ and $\sigma_{xy}$ of triple-layer graphene with the twist angles of -20° (1st and 2nd) and -10° (2nd and 3rd), measured at the temperature of 1.5 K under $B_\perp$ of 2 T, 5 T and 7 T, respectively. **h** PL mapping of the transferred graphene\MoS$_2$ films on Si\SiO$_2$, inset is the typical photograph of the transferred 2-inch graphene\MoS$_2$ vdWS. The horizontal dashed lines in **b**, **e**, **f**, **g** are the guidelines of the filling factors of Landau level, and the dashed lines in **c**, **d** are the guidelines of the magnetic field strength when the Hall plateaus become obvious.

different from half-integer $\nu$ at $\pm 1/2$, $\pm 3/2$ and $\pm 5/3$ for monolayer graphene. Figures 5c and 5d further plot the 2D contour of $R_{xx}$ and $\sigma_{xy}$ under different $B_\perp$ at 1.5 K, and they display the $\nu$ sequence remains robust under $B_\perp$ of large than 2.7 T, indicating the lower QHE threshold for the flat-to-flat transferred double-layer graphene. Extracted $R_{xx}$ and $\sigma_{xy}$ measured at 1.5 K and under $B_\perp$ of 3 T, 5 T and 7 T are shown in supplementary Fig. 13b. In addition, the twist angles of double-layer graphene are considered to renormalize its Landau level[50], so we measure another double-layer graphene film with the twist angle of -13°, which also presents slight p-type doping and the hole mobilities reach -9600 cm$^2$ V$^{-1}$ s$^{-1}$ at 100 K and -6200 cm$^2$ V$^{-1}$ s$^{-1}$ at RT (supplementary Fig. 13c). Figure 5e and supplementary Fig. 13d plot the corresponding $R_{xx}$ and $\sigma_{xy}$ under different measurement conditions. The QHE plateaus also arise at RT, and the Landau filling factors at -7/2, -1, 1, 7/4 and 17/4 can be distinguished approximately at 1.5 K under $B_\perp$ of 7 T, which are significantly different from the double-layer graphene with other twist angles, especially the Bernal AB-stacked bilayer graphene[51]. Figure 5f and supplementary Fig. 13e show more electrical properties of the other double-layer graphene with the twist angel of -20°, with the apparent Landau filling factors at -7/2, -2, -1, 1, 5/2 and 9/2 at 1.5 K under $B_\perp$ of 7 T. Furthermore, the stacking transferred triple-layer graphene films with the twist angles of -20° (1st and 2nd) and -10° (2nd and 3rd) also shows QHE plateaus at 1.5 K under $B_\perp$ up 2 T (Fig. 5g) and with the hole mobilities of -4950 cm$^2$ V$^{-1}$ s$^{-1}$ at 1.5 K (supplementary Fig. 13f). All the robust QHE behaviours of transferred double-layer and triple-layer graphene with controllable twist angles further confirm the high efficiency in fabricating the stacking films with good electrical properties.

The flat-to-flat transfer is also applicable to stack wafer-scale graphene-based vdWS with other 2D materials. Figure 5h shows the photoluminescence (PL) intensity mapping of the monolayer MoS$_2$ films stacked on the transferred graphene film. The inset is the corresponding photograph of the 2-inch graphene\MoS$_2$ on Si\SiO$_2$ wafer, and their typical PL and Raman spectra, Raman intensity mapping for A$_{1g}$ peak of

MoS$_2$ and G peak of graphene, and the AFM at the transition region are shown in supplementary Fig. 14a–f, indicating that the morphological and physical properties of graphene\MoS$_2$ are homogenously distributed over a large area. Moreover, we also fabricate the wafer-sized sapphire\MoS$_2$, sapphire\MoS$_2$\MoSe$_2$ and Si\SiO$_2$\hBN\graphene by the stacking transfer method, as shown in supplementary Fig. 14g–i.

## Discussion

In summary, we develop a flat-to-flat transfer method that combines the spinning-assisted dehydration and PAT processes, and the wafer-scale monolayer, double-layer and triple-layer graphene films are transferred to flat substrates without the formation of cracks, folds and tears. The PAT process can clean the surfaces and interfaces of few-layer graphene, and the doping level can also be reduced. The wetting angle of the solution on graphene is crucial to the transfer process, and the high-quality homogeneous films can be retained while the wetting angle is between 40° to 60°. The twist angles between the double-layer graphene are controlled by adjusting the macroscopic stacking angle between the upper and bottom graphene wafer flats, and the macroscopic stacking angle is calibrated by a high-resolution camera with the angle error of ±1°. Various graphene-based vdWS with specific twist angles are fabricated. The monolayer, double-layer and triple-layer graphene films all show the robust QHE, which even appears at RT with centimetre-sized linewidth. All the transferred graphene-based vdWS present high morphological, structural, optical and electrical homogeneity over the wafer scale, indicating the high efficiency in the stacking transfer method. We believe this transfer method for stacking more graphene-based vdWS will lay the material foundation and accelerate their functional device applications in the near future.

## Methods
### Growth of graphene

Ultra-flat graphene films are grown by the proton-assisted CVD methods as reported before[21], typical parameter is as follows:

sputtered 800 nm Cu-Ni(111) films on c-plane sapphire, growth temperature of 650 °C, pressure of 6 Pa, CH$_4$/H$_2$ ratio of 1:20, plasma power of 15 W and growth time of 5 min. The wrinkled graphene films are grown in a tube furnace by the traditional CVD method, typical parameter is as follows: sputtered 800 nm Cu-Ni(111) films on c-plane sapphire and then annealed at 1050 °C under the hydrogen atmosphere for 60 min to flatten the surface. During growth, growth temperature of 1050 °C, CH$_4$/H$_2$/Ar ratio of 0.1:10:500 and growth time of 10 min. To simplify writing, we use Cu(111) instead of Cu-Ni(111) alloy in our descriptions below.

### Growth of other 2D materials

Wafer-sized MoS$_2$ and MoSe$_2$ films are grown by the two-step vapour deposition method, which includes the typical physical vapour deposition (PVD) and CVD processes, typical parameters are as follows: for PVD process, sputtered 0.8 nm Mo films on SiO$_2$/Si substrate; then for CVD process, Mo films and S or Se powders are located in a two-zone tube furnace, Mo films are heated to 750 °C and S powder is heated to 150 °C for forming MoS$_2$ films, Mo films are heated to 750 °C and Se powder is heated to 280 °C for MoSe$_2$. The H$_2$/Ar mixture gas is used as carrier gas and their flow rate is both 100 sccm. The growth times are all 30 min. Wafer-sized hBN films are grown in a three-inch tube furnace and assisted with the proton process, typical growth parameter is as follows: sputtered 800 nm Cu(111) films on c-plane sapphire and then annealed at 1050 °C under the hydrogen atmosphere for 60 min to flatten the surface. During growth, flattened Cu(111) substrate is heated to 650 °C, the set pressure is 1 Torr, H$_2$/Ar ratio is 100:10, ammonia borane is used as the precursor and loaded into a sealed chamber at the upstream of the growth substrate with the constant temperature at 70 °C. The plasma power is 40 W and the growth time is 10 min. The grown MoS$_2$, MoSe$_2$ and hBN films are all polycrystalline, and they all have ultra-flat surfaces.

### Transfer of monolayer graphene and other 2D materials

The as-grown graphene, hBN films on Cu(111)/sapphire and the as-grown MoS$_2$, MoSe$_2$ films on SiO$_2$/Si are spin-coated with double PMMA films (first, 120k MW, 1 wt% in ethyl lactate, 2000 rpm for 1 min; then, 996k MW, 4 wt% in ethyl lactate, 2000 rpm for 1 min) as the protection layer. Then, 1 M (NH$_4$)$_2$S$_2$O$_8$ aqueous solution is used to etch the Cu(111) substrate and 1 M KOH aqueous solution is used to etch the SiO$_2$. The PMMA/2D material, like PMMA/graphene, is then pulled out from the etchant by a clean polyethylene terephthalate (PET) sheet and moved to DI water for cleaning. The cleaning is normally needed to repeat three times. After that, a two-step spinning-assisted process is developed during pasting the graphene or other 2D materials films onto the flat target substrates (like SiO$_2$/Si). The slow spinning rate of 90–150 rpm is used for the first stage with a spinning time of 6 min, and the fast spinning rate of 300–600 rpm is used for the second stage with a time of 10 min. Then, all the pasted PMMA/graphene on the target substrate is backed at 80 °C for 30 min and at 150 °C for 15 min on a hot plate in sequence. Finally, PMMA is removed by acetone, then the film is dried by a gentle nitrogen gas blow. To further clean the transferred graphene films and decouple them from the substrate, vacuum annealing or PAT is performed. The annealing temperature is 450 °C, the annealing time is 40 min and the vacuum degree is less than $5 \times 10^{-5}$ Pa. The PAT to decouple graphene is performed at 450 °C under 6 Pa, the H$_2$ plasma power is constant 15 W and treatment time is usually 5 min, and the PAT to clean hBN, MoS$_2$ and MoSe$_2$ needs the temperature of 450 °C, 350 °C, and 350 °C, respectively.

### Stacking transfer of graphene-based vdWS

During the stacking transfer process, the as-transferred 2D materials like graphene are used as the flat target substrate. A two-step spinning-assisted process which consists of the same transfer process for monolayer graphene is used for stacking the double layers of graphene films. The difference is that the DI water is replaced by mixing IPA or acetic acid in DI water as the transfer solution, which ensures the wetting angle of solution on graphene falls within the range of 40°–60°. For double-layer graphene, the optimal wetting angles for IPA/DI water volume ratio range from 10 vol% to 25 vol%. For hBN/graphene, the range of IPA is from 15 vol% to 25 vol%, and 20 % of IPA is preferred. For MoS$_2$/graphene, the range of IPA is from 10 vol% to 22.5 vol%, and 15 % of IPA is preferred. For MoS$_2$/MoSe$_2$, the range of IPA is from 7.5 vol% to 17.5 vol%, and 12.5% of IPA is preferred. After the removal of PMMA, the PAT is also performed to clean the surface and the interface between the stacked graphene films. In order to avoid the appearance of C-H bonds during the PAT process, the treatment parameter of double-layer and triple-layer graphene is as follows: H$_2$ plasma of 15 W, treatment time of 5 min, constant pressure of 6 Pa and temperature of 550 °C.

### Characterizations

The optical images are captured by the optical microscope (Nikon LV100ND). Large-area photographs and wetting angles are captured by a high-resolution digital camera. AFM measurements are carried out by a Bruker Dimension Fastscan system at tapping mode under ambient conditions. Raman spectroscopy is carried out by a Witec/alpha 300R confocal microscope with 532 nm laser under ambient conditions, and the in situ variable temperature Raman is performed in a temperature-controlled stage (Linkam THMS600) with introducing pure Ar. All the laser power is set below 2 mW to avoid heating. SAED is performed by a transmission electron microscope (TECNAI F20) with an accelerating voltage of 200 kV, and the sizes of the selected area are all chosen to be ~1 μm. LEED (OCI, BDL600IR-MCP) is measured using electron beam energy of 170 eV at RT under an ultrahigh vacuum of ($1 \times 10^{-8}$ Pa), and the probe electron beam size is about 1 mm.

### Transport measurements

The electrical transport measurements are performed in a $^4$He cryostat with a superconducting magnet (Oxford Teslatron 8T and lowest temperature 1.5 K) with source metre (Keithley 6430) and lock-in amplifier (Stanford SR830). All the transport data are transformed by tensor inversion. The field effect mobility is extracted by $\mu_{FE} = C^{-1}(d\sigma/dV_g)$, where C is the gate capacitance per unit area and calculated as $1.21 \times 10^{-4}$ F/m$^2$ for 285 nm SiO$_2$ layer.

### Data availability

The Source Data underlying the figures of this study are available at https://doi.org/10.6084/m9.figshare.23598834. All raw data generated during the current study are available from the corresponding authors upon request.

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

## Acknowledgements

We thank H.F. Ding, G. Chen & H. N. for their help with LEED experiments, thank Y.M. Dai & X.Q. Zhou for their help with UV-Vis experiments. This study is supported by the National Key R&D Programme of China (Grants Nos. 2021YFA1400400 & 2022YFA1402502), the National Natural Sci-ence Foundation of China (Nos. 12104218 and 51972163), the Funda-mental Research Funds for the Central Universities (Nos. 020414380201 and 020414380176), the Fok Ying-Tong Education Foundation of China (No. 171038) and the China National Postdoctoral Programme for Inno-vative Talents (BX2021120).

## Author contributions

L.G. conceived and supervised the project. L.G. and G.Y. designed the experiments. G.Y. performed graphene transfer, growth, AFM and Raman. W.L. and X.H. performed the device fabrication and transport measurements. Z.W. and B.Y. assisted in the transfer process. C.W. assisted in the SAED measurements. W.S. and Y.N. performed the LEED measurements. K.Y. assisted in the variable temperature Raman mea-surements. H.Z. and Z.Z. assisted in the hBN and TMDC growth. J.X. assisted in the graphene growth and AFM measurements. L.G. and G.Y. analysed the data and wrote the manuscript, J.X. revised it, and all authors commented on it.

## Competing interests

The authors declare no competing interests.
