## [Peer Review File · Nature Communications]

Stacking transfer of wafer-scale graphene-based van der Waals superlatticesEditorial Note: Parts of this Peer Review File have been redacted as indicated to remove third-party material where no permission to publish could be obtained.

REVIEWER COMMENTS

Reviewer #1 (Remarks to the Author):

Review Stacking transfer of wafer-scale graphene-based van der Waals superlattices

The authors have successfully devised a spin-coating-assisted methodology to facilitate the pristine transfer and stacking of two-dimensional (2D) materials. Remarkably, the graphene specimens manipulated via the proposed spin-coating-assisted approach demonstrate a robust quantum Hall effect on a centimeter scale. Furthermore, the authors illustrate that the aforementioned method allows for the precise manipulation of twist angles and stacking configurations across a diverse range of 2D materials. This pivotal accomplishment paves the way for scalable, uncontaminated, and efficient integration of 2D materials.

However, there are several minor inquiries and observations that warrant further clarification and discussion.

1. Page 1 Paragraph 1: The term "spin" commonly refers to the angular momentum of elementary particles within the physics and electrical engineering research communities. To circumvent potential confusion, it is recommended that the authors utilize alternative terminology such as "spin-coating-assisted" or "spinning-assisted" when referring to their methodology.
2. Page 3 Paragraph 2: The authors should provide a succinct and unambiguous definition of the PAT method, notwithstanding its description in the Methods section.
3. Page 4 Paragraph 2: A gentle nitrogen gas (N₂) blow is frequently employed to eliminate bubbles and water during the transfer process (Nanoscale 7.45 (2015): 19099-19109). The authors ought to conduct an investigation comparing the spin-coating-assisted and N₂-gun-assisted methods.
4. Page 7 Paragraph 2: The theoretical size limit of the Quantum Hall Effect (QHE) should be explicitly stated by the authors.
5. Page 9 Paragraph 1: The isopropyl alcohol (IPA)/water mixture has proven effective in multiple previous transfer techniques (e.g., Nanoscale 12.20 (2020): 10890-10911, Matter

4.10 (2021): 3339-3353). The authors have identified graphene wettability as a crucial factor in their spin-coating-assisted stacking. Even though the authors provide a figure in SI, the optimal IPA/water ratio should be mentioned in the main text or Methods section.

6. Page 9 Paragraph 2: The cleaning effect facilitated by the PAT process should be discussed or relevant published results should be cited.

7. Page 11 Paragraph 1: The authors should elaborate on how they control the macroscopic twisting angle and specify the minimum twisting angle achievable using the proposed method.

8. Page 13 Paragraph 1: Although single crystalline graphene was employed in the study, the crystallinity of hBN, MoS₂, and MoSe₂ remains unspecified. The authors should clarify the range of IPA/water ratios that enable the transfer and discuss whether grain boundary intensities affect the final yield of the proposed transfer method.

The publication of the manuscript is recommended upon the completion of revisions, as it advances the development of wafer-scale, ultra-clean transfer of 2D materials.

Reviewer #2 (Remarks to the Author):

In their study, Yuan et al. report a method for transferring graphene layer-by-layer while controlling the interlayer twist angle on a wafer-scale. The researchers introduce a two-step method involving spin-assisted dehydration and proton treatment to remove the aqueous solution under the transferred graphene. As a result, they observe the quantum Hall effect in graphene devices with centimeter-scale dimensions, demonstrating the homogeneity of the film. Overall, the results are promising; however, in my view, the authors need to provide a more detailed explanation of how their method represents a technological advance over previous works to be published in Nature Communications. Detailed comments are provided below.

(1) First of all, proton-assisted treatment (PAT) builds upon previous work (Nature 577, pages 204–208 (2020)), in which a similar quantum Hall effect was observed in a large-scale graphene channel of several hundred micrometers. Although the device size is larger in the current study, the novelty is diminished. Also, while the two-step spin-assisted dehydration is a new method for transferring graphene, the authors need to present the advanced

features more clearly in comparison to previous methods for producing wrinkle-free, clean graphene (Nature Communications volume 13, Article number: 4409 (2022), Nature Communications volume 13, Article number: 5410 (2022)).

(2) While it is noted that there is still a challenge to freely stack homogeneous graphene-based vdWS on a large scale with control of the twist angle, the existing challenges need to be specified in greater detail, as there have been several works published on large-scale stacking of graphene multilayers with controlled twist angles (Nature volume 605, pages 63–68 (2022), Nano Lett. 2022, 22, 4, 1518–1524, Adv Mater 28, 8177-8183 (2016)).

(3) Based on current data, it is not clear whether the twist angle can be controlled, as it is unclear if the 38° degree twist angle shown in Fig. 4 was intentionally targeted. Multilayer graphene with the number of layers > 3 and a constant twist angle would be a good example to prove the controllability. Additionally, the authors need to provide evidence of uniform interlayer coupling between the stacked layers with emerging physical properties to properly motivate the importance of controlling the twist angle. I am particularly concerned about impurities on the top surface of graphene after removing the PMMA supporting layer (Nano Lett. 1, 414419, (2012)), as these impurities could prevent interlayer coupling. While the current technique mostly focuses on removing impurities from the bottom surface of graphene, the authors need to clarify if the hydrocarbon contaminants on the top surface of graphene can also be removed from the center of the film toward the edge using the same process.

Reviewer #3 (Remarks to the Author):

The manuscript by Yuan et al. introduces a novel transfer technique to stack graphene and other 2D materials which allows the realization of wafer-scale van der Waals superlattices. The technique itself combines two distinct steps: a spin-assisted dehydration process and a proton assisted treatment (PAT).

Characterization results indicate that the method is particularly successful at i) transferring and stacking 2D materials on substrates without the formation of cracks, folds and tears, as well as ii) eliminating transfer residues (polymer, water,...). As such, interfaces between 2D

materials or between the bottom 2D material and the substrate are clean, and the assembled van der Waals superlattices are of high-quality (presenting highly morphological, structural, optical and electrical homogeneity over the wafer scale). Finally, authors show a certain degree of control over the twist angle existing between consecutive layers of stacked 2D materials, and thus the technique enables the fabrication of both, conventional and moiré superlattices.

The subject matter of this work is interesting. Superlattices based on van der Waals 2D materials exhibit a wide number of outstanding mechanical, optical and electronic properties, distinct from those of the composing individual 2D materials. Such novel properties are expected to be used for a wide range of existing and emerging applications and the possibility of fabricating these heterostructures with high quality at a wafer-scale is key for the industrial adoption of these materials. I highlight the fact that, although there are a few methods reported in literature to stack 2D materials with controllable twist angle at a large-scale, most of them are either focused on stacking just two graphene layers and/or the studies do not demonstrate the quality of these films over large (cm) scales (see Refs. 4-6 of the article or [Han et al. Nano Lett. 20,3925, 2020]). Furthermore, the study is solid (authors use a wide-range of characterization techniques including Raman spectroscopy, magneto-transport, AFM or LEED to support their claims) and the manuscript reads well.

However, I find some issues that should be addressed before I can recommend this manuscript for publication. They are the following:

- 1.- Authors indicate in the manuscript that the PAT treatment has two effects: i) to clean the transferred films and ii) to decouple the monolayer from the Si/SiO₂ substrate.

I find that results depicted in Supplementary Fig. 3 and 7 are a convincing indication that the permeated protons play an important role in the interface cleaning. Also, results in supplementary Fig. 4 support the fact that protons do not damage the graphene layer (low I(D)/I(G) Raman ratio). Nonetheless, I believe authors should explain further the reason why the coupling between graphene and the substrate is weakened. For instance, is it because of the formation of bonds between the hydrogen atoms and the SiO₂ surface? Is it because of

the production and trapping of re-bonded H₂ at the interface?

2.- Regarding the procedure to control the twist angle. All images in Fig.4 are taken after the slow spin process. I find unclear the way authors adjust/control the so-called “macroscopic angle” α_2 existing between two graphene layers during the pasting operation (step 1 of the transfer technique, prior to the slow spin). As I understand, at this initial stage, the two graphene layers are separated by a thick film of water/IPA and the manipulation/rotation of the top graphene/PMMA film does not seem trivial in such conditions. Moreover, how precise is this adjustment? (i.e. what is the absolute error in the angle when attempting to target a specific α_2 at step 1?). All these details should be included in the text. A picture of the double-layer graphene film at that stage of the process or a video showing the alignment prior and during the two spin processes may notably help to clarify this comment.

3.- Authors mention (methods, transfer section) that “the PMMA/2D material, like PMMA/graphene, is then pulled out by a polyethylene terephthalate (PET) sheet and cleaned by DI water for three times”. For clarity, authors should explain further this step. In particular, does the PET bond on top of the PMMA/2D material floating on the aqueous solution? If so, do authors perform any surface treatment to enhance the bonding with PET?

4.- The temperature at which the PAT process is executed seems to vary depending on whether the 2D material is transferred on the growth substrate (400 Celsius), or stacked on top of another 2D material (550 Celsius). Are these temperatures optimized for the two aforementioned transfer cases? Moreover, I wonder if the PAT technique is able to clean interfaces when processes are run at lower temperatures, including room temperature (e.g. by increasing the processing time). This information is useful to understand the versatility of the technique presented in this study.

5.- Other (minor) comments

- Missing references. The article [Han et al. Nano Lett. 20,3925, 2020] reports a method for the assembly of large-scale 2D transition metal dichalcogenides (TMDs) with controlled layer orientation. As such, it should be included when introducing the topic in the main text

and/or abstract.

- Fig. 2e shows a histogram and Raman map of the peak ratio $I(2D)/I(G)$. For clarity, authors should explicitly indicate (in the main text and caption) if such data corresponds to a sample after the PAT treatment. I note that neighboring panels d and f show three different transfer techniques (as-transferred, annealing and PAT). In this sense, it would be even clearer if authors show in Fig.2e statistical distributions of the $I(2D)/I(G)$ ratio for such 3 processes: as-transferred, annealing and PAT.

- In page 7, authors state “The QHE for the PAT graphene film is the most robust, and the values of integer plateaus are the most accurate and their quantities are also the maximum.”. The expression “ and their quantities are also at the maximum” sounds imprecise. I would suggest to re-write this sentence and use another formulation such as e.g. “and plateaus are fully developed”

- In page 8, authors state “The robust QHE plateaus under relatively mild conditions...”. Could authors explain what do they mean by “relatively mild conditions”?

- In page 10, authors state that “that means the double-layers prefer the 1 – 2 nm periodicity of Moiré patterns, as shown in Fig. 3f.” However, authors do not directly plot the periodicity of the resulting Moiré patterns in Fig.3f but the twist angle between the two graphene layers. Although both parameters are univocally related, (for consistency/clarity reasons) I think authors should select one of the two for both main text and figure.

Referee #1 (Remarks to the Author):

Review Stacking transfer of wafer-scale graphene-based van der Waals superlattices. The authors have successfully devised a spin-coating-assisted methodology to facilitate the pristine transfer and stacking of two-dimensional (2D) materials. Remarkably, the graphene specimens manipulated via the proposed spin-coating-assisted approach demonstrate a robust quantum Hall effect on a centimeter scale. Furthermore, the authors illustrate that the aforementioned method allows for the precise manipulation of twist angles and stacking configurations across a diverse range of 2D materials. This pivotal accomplishment paves the way for scalable, uncontaminated, and efficient integration of 2D materials. However, there are several minor inquiries and observations that warrant further clarification and discussion.

The publication of the manuscript is recommended upon the completion of revisions, as it advances the development of wafer-scale, ultra-clean transfer of 2D materials.

Reply: We thank the Referee #1 for his/her positive comments, strong support and favourable recommendation for our work.

(1) Page 1 Paragraph 1: The term “spin” commonly refers to the angular momentum of elementary particles within the physics and electrical engineering research communities. To circumvent potential confusion, it is recommended that the authors utilize alternative terminology such as “spin-coating-assisted” or “spinning-assisted” when referring to their methodology.

Reply: Thanks for your professional suggestion.

To avoid the unnecessary confusion, we have already used the term “*spinning-assisted*” to replace all “*spin*” for describing our transfer method in the revised manuscript.

(2) Page 3 Paragraph 2: The authors should provide a succinct and unambiguous definition of the PAT method, notwithstanding its description in the Methods section.

Reply: We appreciate your kind advice.

We have already supplemented the description for the simple definition of PAT method in the revised manuscript in Page 3, Paragraph 2 as follows,

“...which consists of the spinning-assisted process and the proton-assisted treatment (PAT) process which is based on the permeated protons and recombined hydrogen, to exhaust the trapped water and decouple the transferred graphene...”

(3) Page 4 Paragraph 2: A gentle nitrogen gas (N₂) blow is frequently employed to eliminate bubbles and water during the transfer process (*Nanoscale* 7.45 (2015): 19099-19109). The authors ought to conduct an investigation comparing the spin-coating-assisted and N₂-gun-assisted methods.

Reply: Thanks for your kind reminder.

It is true that N₂ gas blow is frequently used during the process of transferring graphene, and there are many papers to use this technique, such as (*Nat. Commun.* 10, 867, 2019; *Nat. Commun.* 8, 14560, 2017) as well as your mentioned one (*Nanoscale* 7, 19099-19109, 2015).

During the transfer process, the N₂ gun blow is frequently utilized after the graphene has been pasted on the target substrate with the strong bonding, particularly after removing PMMA. This process is useful for removing the surface residues, such water, acetone, IPA, and others. Actually, this process becomes the standard process through transferring graphene in the wet manner.

We have also adopted the N₂ gun blow to dry graphene films after removing PMMA with acetone and IPA in this study. However, the spinning-assisted method here is designed to uniformly remove the aqueous solution between the PMMA/graphene and target substrate and effectively avoid the consequent cracks and folds.

In the revised Methods (page 16), we have already revised the sentence “... PMMA is removed by acetone, then the graphene film is dried by a gentle nitrogen gas blow.”

Furthermore, to clarify the difference between the two processes, we have also supplemented the following sentence in Page 3, Paragraph 1 as follows,

“Once graphene is firmly attached to the substrate, it is difficult to reduce the trapped water, NPs and consequent folds through the process like nitrogen gas blow^{31, 32, 42}.”

Moreover, we have also supplemented the related papers of (*Nanoscale* 7, 45, 2015; *Nat. Commun.* 10, 867, 2019; *Nat. Commun.* 8, 14560, 2017) as the references 31, 32, and 42.

(4) Page 7 Paragraph 2: The theoretical size limit of the Quantum Hall Effect (QHE) should be

explicitly stated by the authors.

Reply: Thanks for your kind suggestion.

After wide investigations, we find that there is no clear theoretical prediction for the maximum size limit of the quantum Hall effect (QHE), including 2D gases in graphene and GaAs/InGaAs. The realization of QHE can be attributed to several factors, as follows,

- 1) Sample quality. For graphene, an extremely high-crystalline-quality flake or film is needed. Impurities and defects in graphene will cause the electron scattering and strongly affect transport characteristics, but they easily arise in the larger-sized graphene films, which raise the difficulty of QHE with the large linewidth.
- 2) Temperature. Lower temperature helps to reduce the scattering and collisions between electrons and defects, thus improving the electron transport within the Hall devices and narrowing the distribution range of electron levels. On the contrary, higher temperature will also raise the difficulty of QHE.
- 3) Magnetic field. Under stronger magnetic fields, electron energy levels will split into numerous Landau levels and form a discrete energy band structure. Electrons are confined to the Landau levels and can only move along the direction of the magnetic field. Therefore, weaker magnetic fields will also raise the difficulty of QHE.

There are very few reports to realize the QHE for the graphene with the linewidth of more than 100 μm , and they both need the stronger magnetic field and lower temperature, shown in Figure R1 below.

[PANEL B REDACTED]

Figure R1. a, QHE of epitaxial graphene on SiC, which do not need the transfer process. Redrawn from (*Nat. Commun.* 6, 6806, 2015). **b**, QHE of CVD graphene transferred on SiO₂/Si, which needs lower temperature of <0.3 K and stronger magnetic field of >19 T. Redrawn from (*Phys. Rev. B* 90, 115422, 2014).

In our study, we can realize the QHE of graphene with the linewidth up to 1 cm under the near room temperature. It should be the first report of graphene QHE with the linewidth above 0.5 mm, indicating the highly homogenous and high-quality graphene films after our developed transfer method.

(5) Page 9 Paragraph 1: The isopropyl alcohol (IPA)/water mixture has proven effective in multiple previous transfer techniques (e.g., *Nanoscale* 12.20 (2020): 10890-10911, *Matter* 4.10 (2021): 3339-3353). The authors have identified graphene wettability as a crucial factor in their spin-coating-assisted stacking. Even though the authors provide a figure in SI, the optimal IPA/water ratio should be mentioned in the main text or Methods section.

Reply: Thanks for your kind reminder.

We have already supplemented the description in Methods section (page 17) as follows,

“The difference is that the DI water is replaced by mixing IPA or acetic acid in DI water as the transfer solution, which ensures the wetting angle of solution on graphene falls within the range of 40° – 60°. For double-layer graphene, the optimal wetting angles for IPA/DI water volume ratio range from 10 vol% to 25 vol%.”

(6) Page 9 Paragraph 2: The cleaning effect facilitated by the PAT process should be discussed or relevant published results should be cited.

Reply: Thanks for your kind reminder.

The cleaning effect of PAT can be distinguished from the surface of the as-transferred monolayer graphene. After PAT, the monolayer graphene film become clean and flat, as discussed in page 6 paragraph 1 and shown in Fig. 1d and supplementary Fig. 3. We have redrawn some in Figure R2a-b below.

During the stacking transfer, the as-transferred graphene film with flat and clean surface is as the target substrate. Therefore, like the process of transferring monolayer graphene on SiO₂/Si, there are still some PMMA residues on the stacked graphene film even after vacuum annealing, as shown in

Figure R2c. After the PAT process, the surface of stacked double-layer graphene becomes cleaner, as redrawn in Figure R2d. We should be the FIRST ONE to use the PAT process to clean the graphene surface.

Therefore, in Page 9 paragraph 2, we directly use the above results to prove the effectiveness in cleaning graphene surface. Furthermore, we also find that PAT can also clean the interfaces between graphene and substrate as well as double-layer graphene, and we mainly focus on the description of the interface cleaning effect in this paragraph 2 of page 9.

To minimize the confusion, we have already revised the first sentence of this paragraph as follows,

“After spinning-assisted transfer, the stacked graphene film is also treated by PAT process. Typical AFM images of the stacking transferred double-layer graphene are shown in Fig. 3c and supplementary Fig. 7a, and their surfaces are flat and clean, and the thickness ...”

Figure R2. **a**, Typical AFM image of as-transferred graphene film on SiO₂/Si after vacuum annealing. **b**, Cleaning effect of the PAT process for transferred monolayer graphene. Redrawn from supplementary Fig. S3. **c**, Typical AFM image of transferred double-layer graphene film after vacuum annealing. **d**, Cleaning effect of the PAT process for transferred double-layer graphene Redrawn from main Fig. 3c.

(7) Page 11 Paragraph 1: The authors should elaborate on how they control the macroscopic twisting angle and specify the minimum twisting angle achievable using the proposed method.

Reply: Thank for your constructive and professional comment.

1) About control the macroscopic twist angle

A homemade operation system for stacking transfer is used to control the macroscopic twisting angle (Figure R3a), which is pending patent. Figure R3b illustrates the basic control principle. During the staking transfer process, the bottom graphene on SiO₂/Si is secured on the substrate holder and then immersed in the IPA/water solution. Firstly, the bottom graphene is set to a given direction, commonly parallel to a ruler (Figure R3c). The top graphene floating on solution is fixed using a fixer made of two semi-circular shapes (Figure R3d).

To achieve the desired macroscopic angle α_1 between the top and bottom graphene films, we rotate the floating top graphene and calibrate the macroscopic angle according to a high-resolution camera. Once the desired angle has been determined, the top graphene is secured with the fixer (Figure R3e). In principle, the angle α_1 could be set to any angle. The IPA/DI water solution is then slowly exhausted to descend the floating PMMA/graphene, resulting in an initial alignment between the bottom and top graphene films (Figure R3f). After removing the fixer, the aqueous solution between the two graphene films is uniformly removed using the spinning-assisted transfer process (Figure R3g and R3h).

Figure R3. Controlling the macroscopic stacking angle. **a**, Photograph of the customized transfer operation system, consisting of a water basin and a rotation controller. The as-transferred graphene (bottom graphene) is placed on the substrate holder, the rotation controller is used to fix the top graphene, which floating on the aqueous solution. **b**, Schematic of the angle controlling principle, the macroscopic stacking angle is regulated by the wafer flats of graphene. **c**, Photograph of the bottom graphene immersed in the solution with the adjusted wafer flat parallel to the ruler. **d**, Top graphene floating on solution without any control. **e**, Fixed top graphene and adjusting their twist angle α_1 , here α_1 is set at $\sim 20^\circ$. **f**, Exhaust the solution and descend the top-graphene, then finishing an initial alignment between the two graphene films. **g**, Double-layer graphene films after removing the fixer and before spinning, the macroscopic twist angle remains unchanged, $\alpha_2 = \alpha_1 = \sim 20^\circ$. **h**, Double-layer graphene films after spinning-assisted transfer process, the top graphene is nearly not rotated during the spinning process and the final macroscopic twist angle α_3 still remain unchanged. Inset is the triple-layer graphene films after removing PMMA.

2) About the angle error

In this study, the measurement and calibration of twist angles are performed with the assistance of a high-resolution camera, which has the angle resolution of $\sim 1^\circ$. The twist angle remains unchanged within the resolution range during slowly exhausting the aqueous solution to achieve the initial

alignment (Figure R3f) and after moving away the fixer (Figure R3g, also known as “Before spinning”). Furthermore, after the spinning transfer process (Figure R3h), the final obtained angle α_3 still remains unchanged, indicating the top PMMA/graphene film is firmly attached in place during the spinning process.

The operation to maintain the floating graphene in place should involve two crucial factors:

1) The aqueous solution is exhausted until it is beneath the top graphene before spinning, and there is little or no solution out of graphene region. Only in this way, PMMA/graphene will not move laterally with the solution.

2) During the low spinning process, the aqueous solution is exhausted from the inner to the edge, and the top PMMA/graphene is unable to rotate compared to the bottom graphene. This should be attributed to the van der Waals interaction between the top and bottom graphene, which prevent the top graphene to rotate.

Consequently, α_3 is consistently measured to be equal to α_2 , which is also equal to α_1 . In Figure R3, we demonstrate the ability to control angle $\alpha_3 = \alpha_2 = \alpha_1 = 20^\circ (\pm 1^\circ)$.

3) About the obtained minimum twist angle

Figure R4 shows another transferred double-layer graphene with the designed twist angle of roughly 0° . The twist angle of $\alpha_3 = \alpha_1$ for the final films remains 0° measured by the camera. Noting that, a higher resolution optical image shows that the twist angle is about $\sim 1^\circ$ according to the two wafer-flats (Figure R4f). Thus, we could make sure the minimum twist angle of about $0^\circ (\pm 1^\circ)$.

Figure R4. Controlling the macroscopic stacking angle of 0° . **a**, Photo of the bottom graphene immersed in aqueous solution and adjust the wafer-flat of graphene parallel to the ruler. **b**, Top PMMA/graphene floating on aqueous solution without any control. **c**, Adjusting the macroscopic twist angle at approximately 0° . **d**, Exhaust the solution to descend the top-graphene and finally achieve an initial alignment between the two graphene films. **e**, Graphene films after spinning-assisted transfer process, photo captured by camera showing that the as-obtained twist angle still remains $\sim 0^\circ$. **f**, Higher resolution optical image shows that the twist angle is about 1° .

To further demonstrate the controllability and repeatability, we have transferred a triple-layer graphene films with the regulated angles (Figure R5). Firstly, another double-layer graphene films with the twist angle of roughly 20° is transferred, then we transfer a third graphene film with the twist angle of 0° (between the second and the third graphene films). After the stacking process, we perform TEM for the transferred triple-layer graphene films, and the SAED patterns show the twist angles are also nearly identical to the macroscopic angles.

Figure R5. Stacking transfer of triple-layer graphene with controlled twist angles. **a**, Photo of the as-transferred double-layer graphene films with macroscopic stacking angle α_1 of approximately 20° , immersing the film in the aqueous solution and adjusting one of the wafer-flat of graphene films parallel to the ruler. **b**, Adjusting the macroscopic stacking angle α_2 at $\sim 0^\circ$ (i.e. $\sim 20^\circ$ to another graphene). **c**, Exhausting the solution, descending the top graphene to finally achieve an initial alignment between the layers. **d**, Graphene films after spinning-assisted transfer process, and zoom-in optical image shows that the twist angles of $\sim 0^\circ$ and $\sim 20^\circ$. **e**, SAED patterns of the triple-layer graphene collected at random locations, the twist angles are basically consistent with the macroscopic stacking angle.

In revised manuscript, we have supplemented Figure R3 and R4 as the new supplementary Fig. 10 and 11, and Figure R5a-c and part of Figure R5d-e are also supplemented as the new Fig. 4b-e in revised manuscript. The original Fig. 4b-d are moved as the supplementary Fig. 12a-c. The angle control process is further re-written in detail on page 12 and page 13 of the revised manuscript.

(8) Page 13 Paragraph 1: Although single crystalline graphene was employed in the study, the crystallinity of hBN, MoS₂, and MoSe₂ remains unspecified. The authors should clarify the range of IPA/water ratios that enable the transfer and discuss whether grain boundary intensities affect the final yield of the proposed transfer method.

Reply: Thank for your constructive and professional comment.

1) Crystallinity of hBN, MoS₂ and MoSe₂

In this study, hBN films are grown by the proton-assisted CVD method and they are polycrystal

with an **ultra-flat surface** (Figure R6a-b). Moreover, the polycrystalline hBN films present a good breakdown voltage of ~ 1.8 V for the thickness of ~ 2 nm (Figure R6c). Notably, the mostly CVD grown hBN films cannot be used as the dielectric material because the electric leakage even for the single-crystalline hBN multi-layers (*Nature* 606, 88-93, 2022; *Nat. Electron.* 6, 126-136, 2023). We are also **preparing another paper** for growing the insulating hBN film, and it will be submitted within two months.

MoS₂ and MoSe₂ are grown by our developed two-step vapor deposition method (*Nat. Mater.* 18, 602-607, 2019), and the method is firstly used for growing environmental stable NbSe₂ and TiSe₂. In spite of polycrystal, NbSe₂ films still exhibit good superconducting behaviour and excellent environmental stability (Figure R6d-e). Similarly, the obtained MoS₂ and MoSe₂ grown with this method are also polycrystalline, a typical STEM image of MoS₂ is shown in Figure R6f. Note that, the as-grown MoS₂ film also have **ultra-flat surface** (Figure R6g) and this feature actually shows a crucial role in the subsequent stacking transfer in a flat-to-flat manner.

It is worth noting that the **flat surface** is critical for our flat-to-flat transfer process and the subsequent stacking transfer process. Furthermore, only the single crystalline films can be used to manipulate the twisting angle between the top and bottom 2D materials. Therefore, we do not emphasize the twisting angle between graphene and MoS₂, MoSe₂ and hBN until the single crystalline films with flat surface can be grown.

Figure R6. **a**, Typical SAED pattern of the 2-nm-thick hBN films, the films are polycrystalline. **b**, Typical AFM image of the transferred hBN films, the films have the ultra-flat surface. **c**, Typical current density–voltage curves for 2-nm-thick hBN film. **d**, STEM image of the NbSe₂ films show that the films are polycrystalline. **e**, Robust superconductivity of bilayer NbSe₂ after vacuum annealing for 60 min (red line) and being immersed in different aqueous solutions for 30 min (blue line). **d** and **e** are redrawn from (*Nat. Mater.* 18, 602-607, 2019). **f**, STEM image of the MoS₂ films show that the films are polycrystalline. **g**, Typical AFM image of the MoS₂ films, the film has the ultra-flat surface.

To minimize the confusion, we have already supplemented the description in the “Growth of other 2D materials” of Methods section as follows,

“...The grown MoS₂, MoSe₂ and hBN films are all polycrystalline, and they all have the ultra-flat surfaces.”

2) About the optimum range of IPA/water for stacking transfer

In revised manuscript, we have supplemented the following descriptions for the optimum range of IPA concentration for different stacking transfer in the “Stacking transfer of graphene-based vdWS” of Methods section as follows,

“... For hBN/graphene, the range of IPA is from 15 vol% to 25 vol%, and 20 vol% of IPA is preferred. For MoS₂/graphene, the range of IPA is from 10 vol% to 22.5 vol%, and 15 vol% of IPA is preferred. For MoS₂/MoSe₂, the range of IPA is from 7.5 vol% to 17.5 vol%, and 12.5 vol% of IPA is preferred...”

Referee #2 (Remarks to the Author):

In their study, Yuan et al. report a method for transferring graphene layer-by-layer while controlling the interlayer twist angle on a wafer-scale. The researchers introduce a two-step method involving spin-assisted dehydration and proton treatment to remove the aqueous solution under the transferred graphene. As a result, they observe the quantum Hall effect in graphene devices with centimeter-scale dimensions, demonstrating the homogeneity of the film. Overall, the results are promising; however, in my view, the authors need to provide a more detailed explanation of how their method represents a technological advance over previous works to be published in Nature Communications. Detailed comments are provided below.

Reply: We thank the Referee #2 for his/her positive comments, strong support and favourable recommendation for our work.

(1) First of all, proton-assisted treatment (PAT) builds upon previous work (Nature 577, pages 204–208 (2020)), in which a similar quantum Hall effect was observed in a large-scale graphene channel of several hundred micrometers. Although the device size is larger in the current study, the novelty is diminished. Also, while the two-step spin-assisted dehydration is a new method for transferring graphene, the authors need to present the advanced features more clearly in comparison to previous methods for producing wrinkle-free, clean graphene (Nature Communications volume 13, Article number: 4409 (2022), Nature Communications volume 13, Article number: 5410 (2022)).

Reply: Thank for your constructive and professional comment.

1) Difference from the previous QHE

In our previous study (Nature 577, 204-208, 2020), we have developed a new growth method, called proton assisted CVD, to grow the ultra-flat graphene films. However, the graphene films on Cu(111) are still transferred to insulating substrates through the traditional wet transfer methods. We find that the electrical properties of the transferred film still have a huge room for improvement based on the results of the QHE and the carrier mobility.

For instance, although we observed the QHE in the Hall devices over a hundred-micron linewidth,

the longitudinal resistivity (R_{xx}) approached 0Ω only in the first filling factor ($\nu = \pm 2$). Starting from the second filling factor with $\nu = \pm 6$, the R_{xx} fails to reach 0Ω due to the weakly quantized Landau level. The Hall conductivity (σ_{xy}) also shows the imprecise plateau from $\nu = \pm 6$. The QHE features of Hall device with 500-micron linewidth are even worse with the weakly quantized Landau level at $\nu = \pm 2$ (Figure R7a). These imperfect results of Hall devices with more than 500-micron linewidth inspire us to develop a new transfer method to achieve the wafer-scale graphene films on the target substrates with high electrical quality. In this study, we have successfully developed a new flat-to-flat transfer methods and achieve the more robust QHE characteristics even with the centimeter linewidth (Figure R7b). The measurement condition can be more gentle, like higher temperature or weaker magnetic field. The QHE characteristics with the same performances in the exfoliated graphene with micron-sized linewidth are obtained under the much higher magnetic field (Figure R7c).

Additionally, the carrier mobility (μ_{FET}) by the traditional wet transfer method is extracted to be average $4,900 \text{ cm}^2\text{V}^{-1}\text{s}^{-1}$ at room temperature (RT) in our previous work, and the μ_{FET} is increased to be average $9,300 \text{ cm}^2\text{V}^{-1}\text{s}^{-1}$ at RT in this study. They are all measured in the Hall devices with hundred-micron linewidth. Therefore, the new flat-to-flat transfer method can make sure the high uniformity and high quality of the graphene films across the wafer level.

2) Difference from the previous transfer methods

As mentioned in the introduction of manuscript (page 2, paragraph 2), “...it is relatively easy to realize the transfer of wafer-scale monolayer graphene...”. In the previous studies like (*Nat. Commun.* 13, 4409, 2022; *Nat. Commun.* 13, 5410, 2022), the authors successfully realized the high-level transfer of large-area monolayer graphene via the gradient surface energy modulation and controllable conformal contact, respectively. In fact, there are already more than 20 different methods reported to transfer monolayer graphene films, including the bubbling transfer method (*Nat. Commun.* 3, 699, 2012) and the face-to-face transfer method (*Nature* 505, 190-194, 2014) developed by the corresponding author Libo Gao. However, most of these methods primarily focus on transferring monolayer graphene to a different target substrate.

To date, it remains challenging to prepare homogeneous graphene-based van der Waals structures (vdWS) at wafer scale, such as double-layer and triple-layer graphene, as well as wafer-scale graphene

combined with other 2D materials. In this study, we mainly focus on the stacking transfer of homogeneous graphene-based vdWS at wafer scale and further controlling their twist angles (Figure R7d).

Figure R7. **a**, QHE in Hall device with hundred-micron linewidth measured for the ultra-flat graphene transferred by traditional wet transfer method. Redrawn from (*Nature* 577, 204-208, 2020). **b**, Photo of the centimetre-sized Hall bar and measured QHE, the robust Hall plateaus can be easily observed under $B_{\perp} = 7$ T. Redrawn from main Fig. 2h and 2i. **c**, QHE obtained in exfoliated graphene measured at $B_{\perp} = 14$ T and $T = 4$ K. Redrawn from (*Nature* 438, 197-200, 2005). **d**, Photo of homogeneous triple-layer graphene films obtained by sequential stacking 4-inch graphene to a 6-inch SiO_2/Si wafer. Redrawn from main Fig. 3a.

(2) While it is noted that there is still a challenge to freely stack homogeneous graphene-based vdWS on a large scale with control of the twist angle, the existing challenges need to be specified in greater detail, as there have been several works published on large-scale stacking of graphene multilayers with controlled twist angles (*Nature* volume 605, pages 63–68 (2022), *Nano Lett.* 2022, 22, 4, 1518–1524, *Adv Mater* 28, 8177-8183 (2016)).

Reply: Thank for your constructive and professional comment.

As you described, “*There is still a challenge to freely stack homogeneous graphene-based vdWS on a large scale with control of the twist angle*”. We find that the biggest challenges here is “**uniformity**” in large scale. The macroscopical transfer defects are the main troubles for the homogeneous features, like folds, wrinkles, cracks, residual particles and so on. They not only affect the morphological

flatness and the uniformity of the transferred films, but also seriously hinder the subsequent stacking process. If the transferred graphene films with un-flatted and un-cleaned surface act as the underlying substrate, the top graphene film will be easily broken. Therefore, we unanimously agree to that the difficulty of transferring graphene are growing exponentially with the sample scale (*Nat. Commun.* 13, 4409, 2022; *Nanoscale* 12, 10890-10911, 2020; *et al.*).

The stacking transfer for constructing large-area multilayer graphene can be traced back to as early as (*Nano Lett.* 9, 4359-4363, 2009; redrawn in Figure R8a), and there are also several papers to report on the large-scale stacking of multilayer graphene with controlled twist angles. However, most of them does not demonstrate the uniformity and the physical properties over large scale. For example, (*Nature* 605, 63-68, 2022; redrawn in Figure R8b) reported the preparation of twisted double-layer graphene by simply folding one single crystalline graphene grain, and they only measure the plasmon mode within micron scale. (*Nano Lett.* 22, 1518-1524, 2022; redrawn in Figure R8c) and (*Adv. Mater.* 28, 8177-8183, 2016; redrawn in Figure R8d) successfully controlled the twist angles of graphene, but the homogeneity about the macroscopic morphology and the physical qualities at large scale are not investigated. We find that the as-transferred graphene commonly contains some residuals on the surface, which will inevitably affect the further stacking process. There are usually lack of morphological characterizations, like AFM. Although the well QHE is obtained, the measured Hall device is still needed to be covered with hBN flakes and the linewidth is still at micron scale.

By the way, the referee #3 also support this: “*I highlight the fact that, although there are a few methods reported in literature to stack 2D materials with controllable twist angle at a large-scale, most of them are either focused on stacking just two graphene layers and/or the studies do not demonstrate the quality of these films over large (cm) scales*”).

In page 3 paragraph 1 of manuscript, we have emphasized the challenge of the homogeneous features during the transfer process. Furthermore, we have also revised the statement in page 2 paragraph 2:

Changing “...Until now, it is relatively easy to realize the transfer of wafer-scale monolayer graphene, *but there still remains a challenge to freely stack homogeneous graphene-based vdWS with wafer scale, and it is more difficult to control their twist angles between the neighbouring layers.*”

into

“...Until now, it is relatively easy to realize the transfer of wafer-scale monolayer graphene, and there are also a few approaches for manipulating the twist angles between the adjacent graphene layers. However, stacking homogeneous graphene-based vdWS at wafer size remains a challenge, and the homogeneous vdWS with controlled twisted angles is more difficult.”

Furthermore, we have also supplemented the citations for the published papers (*Nano Lett.* 22, 1518-1524, 2022; *Nature* 605, 63-68, 2022) in the revised manuscript.

[PANELS A, C AND D REDACTED]

Figure R8. **a**, Photo of 1 cm² of transferred graphene on glass slips with 1 to 4 layers. Redrawn from (*Nano Lett.* 9, 4359-4363, 2009). **b**, Optical image of the prepared twist double-layer graphene, inset is the schematic of the preparation method from a graphene single crystal, and the SEM image of the tested device with nanoribbon structure. Redrawn from (*Nature* 605, 63-68, 2022). **c**, Typical AFM image of the transferred graphene. Redrawn from (*Nano Lett.* 22, 1518-1524, 2022). **d**, Fabrication of Hall device, where the graphene is protected with thick hBN flakes and the linewidth of the device is micron. Redrawn from (*Adv. Mater.* 28, 8177-8183, 2016).

(3) Based on current data, it is not clear whether the twist angle can be controlled, as it is unclear if the 38° degree twist angle shown in Fig. 4 was intentionally targeted. Multilayer graphene with the number of layers >3 and a constant twist angle would be a good example to prove the controllability. Additionally, the authors need to provide evidence of uniform interlayer coupling between the stacked layers with emerging physical properties to properly motivate the importance of controlling the twist angle. I am particularly concerned about impurities on the top surface of graphene after removing the PMMA supporting layer (*Nano Lett.* 1, 414419, (2012)), as these impurities could prevent interlayer coupling. While the current technique mostly focuses on removing impurities from the bottom surface of graphene, the authors need to clarify if the hydrocarbon contaminants on the top surface of graphene can also be removed from the center of the film toward the edge using the same process.

Reply: Thank for your constructive and professional comment.

1) About the twist angle control

This comment is similar to the comment (7) of Referee #1. A more detailed angle control procedures with a homemade operation system are used. We have replied to this comment above and shown the new results in Figure R3-R5.

In brief, we can freely control the stacking angles between the adjacent graphene films, with the absolute error of $\pm 1^\circ$. In Figure R3 and R4, we have obtained two pieces of wafer-scale double-layer graphene films with twist angles of about 0° and 20° , respectively. To further illustrate the controllability and repeatability, we have also transferred a triple-layer graphene films with controlled angles (Figure R5) through firstly transferring double-layer graphene with twist angle of 20° and further transferring a third graphene film with twist angle 0° . The SAED patterns at random locations show that the macroscopic stacking angles are basically consistent with the designed twist angles.

2) About the physical properties based on the uniform interlayer coupling

In this study, we have performed the large-area Raman mapping of the double-layer and triple-graphene as well as the QHE of Hall devices with linewidth up to hundred microns, which is made by the twisted few-layer graphene films, indicating their uniform interlayer coupling. For the double-layer graphene films with different twist angles, there are different Landau filling factors of QHE plateaus, which result from the renormalization of Landau levels caused by twist angles. Typical transport characteristics of double-layer graphene with twist angles of 10° and 13° are redrawn in Figure R9a-b (Figs. 5b and 5e of manuscript). In this study, we mainly focus on the new stacking transfer method to construct the homogeneous graphene vdWS with wafer scale, and the excellent electrical properties of double-layer graphene fully demonstrate the controllability of the flat-to-flat transfer method and the high uniformity of the transferred films.

We agree to the referee' strong interest in the possible undiscovered physical properties of the few-layer graphene with different twist angles. During the revision period, we have also conducted more electrical measurements for double-layer (20°) and triple-layer ($20^\circ/10^\circ$) graphene films, as shown in Figure R9c and R9d.

Figure R9. **a**, R_{xx} and σ_{xy} of the double-layer graphene with $\sim 10^\circ$ twist angle as a function of V_{bg} under B_\perp of 7 T at 1.5 K, 77 K and RT, respectively. Redrawn from main Fig. 5b. **b**, R_{xx} and σ_{xy} of double-layer graphene with $\sim 13^\circ$ twist angle as a function of V_{bg} under B_\perp of 7 T at 1.5 K, 100 K and RT, respectively. Redrawn from main Fig. 5e. **c**, (upper) R_{xx} and σ_{xy} of double-layer graphene with twist angle of $\sim 20^\circ$, measured under B_\perp of 7 T and at the temperature of 1.5 K 100 K and RT, respectively. (bottom) R_{xx} and σ_{xy} of double-layer graphene with twist angle of $\sim 20^\circ$, measured at the temperature of 1.5 K under B_\perp of 2 T, 5 T and 7 T, respectively. **d**, (upper) R_{xx} and σ_{xy} of triple-layer graphene with twist angle between first and second layers of $\sim 20^\circ$, and twist angle between second and third layers of $\sim 10^\circ$, measured at the temperature of 1.5 K under B_\perp of 2 T, 5 T and 7 T, respectively. (bottom) R_{xx} of the corresponding triple-layer graphene film as a function of V_{bg} at 1.5 K and extracted mobility.

3) About the impurities on the top surface

In this study, we have compared the residual nanoparticles (NPs) on the surface of the as-transferred graphene, graphene after vacuum annealing and after PAT, demonstrating that the annealing process will remove some NPs and the PAT can make graphene surface much cleaner, as shown in supplementary Fig. 3. Furthermore, we have also performed the *ex situ* AFM image of the as-transferred graphene after different treatments, as shown in Figure R9. The results show that no significant movement of surface contaminants during the PAT cleaning process, indicating the residual NPs are directly removed on the graphene surface.

Figure R10. *ex situ* AFM images of the as-transferred graphene film after different PAT process. **a**, As-transferred graphene with a lot of residual NPs on the surface. **b**, After PAT process of 15 W, 5 s. **c**, After PAT process of 15 W, 20 s. **d**, After PAT process of 15 W, 60 s.

In the revised manuscript, we have supplemented Fig. R3 and R4 as the new supplementary Fig 10 and 11, and Figure R5a-c and part of Figure R5d-e are also supplemented as the new Fig. 4b-e in revised manuscript. The original Fig. 4b-d are moved as the supplementary Fig. 12a-c. The angle control process is further re-written in detail on page 12 and page 13 of the revised manuscript.

Furthermore, we have also supplemented the Figure R9c-d and Figure R10 as the main Fig. 5f-g and supplementary Fig. 3c in the revised manuscript.

Referee #3 (Remarks to the Author):

The manuscript by Yuan et al. introduces a novel transfer technique to stack graphene and other 2D materials which allows the realization of wafer-scale van der Waals superlattices. The technique itself combines two distinct steps: a spin-assisted dehydration process and a proton assisted treatment (PAT).

Characterization results indicate that the method is particularly successful at i) transferring and stacking 2D materials on substrates without the formation of cracks, folds and tears, as well as ii) eliminating transfer residues (polymer, water,). As such, interfaces between 2D materials or between the bottom 2D material and the substrate are clean, and the assembled van der Waals superlattices are of high-quality (presenting highly morphological, structural, optical and electrical homogeneity over the wafer scale). Finally, authors show a certain degree of control over the twist angle existing between consecutive layers of stacked 2D materials, and thus the technique enables the fabrication of both, conventional and moiré superlattices.

The subject matter of this work is interesting. Superlattices based on van der Waals 2D materials exhibit a wide number of outstanding mechanical, optical and electronic properties, distinct from those of the composing individual 2D materials. Such novel properties are expected to be used for a wide range of existing and emerging applications and the possibility of fabricating these heterostructures with high quality at a wafer-scale is key for the industrial adoption of these materials. I highlight the fact that, although there are a few methods reported in literature to stack 2D materials with controllable twist angle at a large-scale, most of them are either focused on stacking just two graphene layers and/or the studies do not demonstrate the quality of these films over large (cm) scales (see Refs.4-6 of the article or [Han et al. Nano Lett. 20,3925, 2020]). Furthermore, the study is solid (authors use a wide-range of characterization techniques including Raman spectroscopy, magneto-transport, AFM or LEED to support their claims) and the manuscript reads well.

Reply: We much thank the Referee #3 for his/her positive comments, strong support and favourable recommendation for our work.

However, I find some issues that should be addressed before I can recommend this manuscript for publication. They are the following:

(1) Authors indicate in the manuscript that the PAT treatment has two effects: i) to clean the transferred films and ii) to decouple the monolayer from the Si/SiO₂ substrate.

I find that results depicted in Supplementary Fig. 3 and 7 are a convincing indication that the permeated protons play an important role in the interface cleaning. Also, results in supplementary Fig. 4 support the fact that protons do not damage the graphene layer (low I(D)/I(G) Raman ratio). Nonetheless, I believe authors should explain further the reason why the coupling between graphene and the substrate is weakened. For instance, is it because of the formation of bonds between the hydrogen atoms and the SiO₂ surface? Is it because of the production and trapping of re-bonded H₂ at the interface?

Reply: Thank for your constructive and professional comment.

We believe the decoupling is due to the **re-bonded H₂ at the interfaces** between graphene and the substrate. Unfortunately, the detection of tiny H₂ in the interlayer is difficult for most characterization technologies until now. The Raman and ARPES characterizations confirm the hydrogen intercalated SiC(0001), as shown in Figure R11 (*2D Mater.* 3, 025023, 2016; *Phys. Rev. Lett.* 103, 246804, 2009).

[FIGURE REDACTED]

Figure R11. a, Raman maps of Si-H and 2D (graphene) peak intensity for the quasi-freestanding monolayer graphene on SiC, and individual spectra collected from the marked areas (MNPs) on terraces and edges, showing the enhanced Si-H and 2D peaks. Redrawn from (*2D Mater.* 3, 025023, 2016). **b**, Dispersion of the π bands measured with ARPES for an as-grown monolayer graphene after hydrogen annealing treatment at different temperature. Redrawn from (*Phys. Rev. Lett.* 103, 246804, 2009).

We have adopted high resolution Raman spectroscopy and XPS to detect the re-combined H₂ or H-Si bonds (Figure R12a-b), but there are no apparent signals for H₂ or H-Si bonds.

We have also treated the graphite (or multilayer graphene) by similar PAT processes with a longer treatment time recently. After PAT, we find there are many bubbles and the gas in the bubbles are confirmed as molecular hydrogen (H₂) by Raman spectroscopy (Figure R12c), indicating that the permeated protons are re-combined into H₂. We are also **submitting another study** for the formation of H₂ bubbles now.

We have still treated the epitaxial graphene on SiC(0001) by PAT process. The graphene on SiC is grown by the similar method to those have been reported. ARPES for the graphene on SiC(0001)

before and during PAT processes are shown in Figure R12d. The PAT shows the similar effect of H₂ annealing under atmospheric pressure at higher temperature (>900 °C), as shown Figure R11b. We are also **preparing another study** for the decoupling of graphene grown on SiC recently.

For the sake of caution and to minimize the confusion, we have already revised the corresponding description of PAT method in Page 3, paragraph 2 as follows,

“... which consists of the spinning-assisted process and the proton-assisted treatment (PAT) process **which is based on the permeated protons and recombined hydrogen**, to exhaust the trapped water and decouple the transferred graphene...”

Also in page 5, paragraph 1 as follows,

“...**the permeated protons and re-bonded H₂ should play an important role in decoupling graphene...**”

Figure R12. **a**, Raman spectra of graphene on SiO₂/Si with different PAT processes, there is no apparent H-Si (2,100 – 2,150 cm⁻¹) and H₂ (4,100 – 4,150 cm⁻¹) peaks. **b**, XPS of the transferred graphene with different PAT processes, there are no new peaks and the binding energy of C 1s is not shifted. **c**, Optical and AFM images of the bubbles which are produced by long-term PAT, Raman spectra show that the gases in the bubbles is molecular hydrogen. **d**, ARPES of as-grown graphene on SiC(0001), after PAT and subsequent UHV annealing.

(2) Regarding the procedure to control the twist angle. All images in Fig.4 are taken after the slow spin process. I find unclear the way authors adjust/control the so-called “macroscopic angle” α_2 existing between two graphene layers during the pasting operation (step 1 of the transfer technique, prior to the slow spin). As I understand, at this initial stage, the two graphene layers are separated by a thick film of water/IPA and the manipulation/rotation of the top graphene/PMMA film does not seem trivial in such conditions. Moreover, how precise is this adjustment? (i.e. what is the absolute error in

the angle when attempting to target a specific α_2 at step 1?). All these details should be included in the text. A picture of the double-layer graphene film at that stage of the process or a video showing the alignment prior and during the two spin processes may notably help to clarify this comment.

Reply: Thank for your constructive and professional comment.

This comment is similar to the comment (7) of Referee #1. A more detailed angle control procedures with a **homemade operation system** are used. We have replied to this comment above and shown the new results in Figure R3-R5.

The twist angle of wafer-scale graphene films is measured and calibrated with the assistance of a camera which with an **angle resolution of $\sim 1^\circ$** . The top graphene layer remains stationary and unrotated compared to the bottom graphene during the stacking transfer process, so the twist angle before and after the spinning process remains unchanged (within the margin of error $\pm 1^\circ$), as shown in Figures R4.

The operation to maintain the top graphene in place should involve two crucial factors:

1) The aqueous solution is exhausted until it is beneath the top graphene before spinning, and there is little or no solution out of graphene region (Figure R3g). Only in this way, PMMA/graphene will not move laterally with the solution.

2) During low spinning process, the aqueous solution is exhausted from the inner to the edge, and the top PMMA/graphene is unable to rotate. This should be attributed to the van der Waals interaction between the top and bottom graphene, which prevent the top graphene to rotate.

We have already transferred the wafer-scale twisted double-layer graphene with twist angles of roughly 0° and 20° in Figures R3-R4. Figure R5 shows the transferred triple-layer graphene with regulated angles, further demonstrating the controllability and reproducibility. After the stacking process, we further perform TEM for the transferred triple-layer graphene films, and the SAED patterns show the twist angles are also nearly identical to the macroscopic angles.

In the revised manuscript, we have supplemented Figure R3 and R4 as the new supplementary Fig. 10 and 11, and Figure R5a-c and part of Figure R5d-e are also supplemented as the new Fig. 4b-e in revised manuscript. The angle control process is further re-written in detail on page 12 and page 13 of the revised manuscript.

(3) Authors mention (methods, transfer section) that “the PMMA/2D material, like PMMA/graphene, is then pulled out by a polyethylene terephthalate (PET) sheet and cleaned by DI water for three times”. For clarity, authors should explain further this step. In particular, does the PET bond on top of the PMMA/2D material floating on the aqueous solution? If so, do authors perform any surface treatment to enhance the bonding with PET?

Reply: Thank for your constructive and professional comment.

Our description may cause the ambiguity, so we have revised the related statement in page 16 of manuscript as follows,

“...The PMMA/2D material, like PMMA/graphene, is then pulled out from the etchant by a clean polyethylene terephthalate (PET) sheet and moved to DI water for cleaning. The cleaning is normally needed to repeat three times...”

During our experiment, we do not perform any surface treatment to PET sheet, and PET will not bond with the PMMA/graphene during the short contact time. PET sheet is only used as an intermediate substrate for the purpose of moving the PMMA/graphene from one solution to another, and it only needs the clean surface whatever the surface is hydrophilic or hydrophobic. Actually, the glass slide can also act as this intermediate substrate. In Figure R13, we also demonstrate the processes of moving PMMA/ graphene from solution A (DI water) to solution B (DI water) with PET sheet.

Figure R13. Moving PMMA/graphene from solution A to solution B with PET sheet. **a**, Pulling out PMMA/graphene from solution A. **b**, Moving PMMA/graphene to solution B. **c**, PMMA/graphene floating on solution B.

(4) The temperature at which the PAT process is executed seems to vary depending on whether the 2D

material is transferred on the growth substrate (400 Celsius), or stacked on top of another 2D material (550 Celsius). Are these temperatures optimized for the two aforementioned transfer cases? Moreover, I wonder if the PAT technique is able to clean interfaces when processes are run at lower temperatures, including room temperature (e.g. by increasing the processing time). This information is useful to understand the versatility of the technique presented in this study.

Reply: Thank for your constructive and professional comment.

We have optimized the PAT temperatures through the hydrogenation of 2D materials. Graphene is easy to be hydrogenated and arise Raman D peak (C-H bonds) at low temperature, and most hydrogenated graphene, like graphane or graphone, are prepared by this condition. For monolayer graphene, the PAT temperature should be higher than 350 °C (Figure R14a). For double-layer and triple-layer graphene, the PAT temperature should be higher than 500 °C (Figure R14b-c). In order to avoid the appearance of C-H bonds during the PAT process, we need to choose the PAT temperature higher than the critical temperature.

We are **preparing another study** for the hydrogenated graphene, so it is not appropriate to supplement Figure R14 into the revised edition.

Furthermore, we have already revised the descriptions about the PAT condition in the “Transfer of monolayer graphene and other 2D materials” of Methods section as follows,

“...The PAT to decouple monolayer graphene on SiO₂/Si is performed at 400 °C under 6 Pa, the H₂ plasma power is constant 15 W and treatment time is usually 5 min, and the PAT to clean hBN, MoS₂ and MoSe₂ needs the temperature of 400 °C, 350 °C, 350 °C, respectively.”

and “Stacking transfer of graphene-based vdWS” of Methods section as follows,

“...In order to avoid the appearance of C-H bonds during the PAT process, the treatment parameter of double-layer and triple-layer graphene...”

Figure R14. Raman spectra of monolayer (a), double-layer (b), and triple-layer (c) graphene after PAT at different temperature.

(5) Other (minor) comments: Missing references. The article [Han et al. Nano Lett. 20,3925, 2020] reports a method for the assembly of large-scale 2D transition metal dichalcogenides (TMDs) with controlled layer orientation. As such, it should be included when introducing the topic in the main text and/or abstract.

Reply: Thanks for your kind reminder.

We have already supplemented this related paper as the new reference (9).

(6) Fig. 2e shows a histogram and Raman map of the peak ratio $I(2D)/I(G)$. For clarity, authors should explicitly indicate (in the main text and caption) if such data corresponds to a sample after the PAT treatment. I note that neighboring panels d and f show three different transfer techniques (as-transferred, annealing and PAT). In this sense, it would be even clearer if authors show in Fig.2e statistical distributions of the $I(2D)/I(G)$ ratio for such 3 processes: as-transferred, annealing and PAT.

Reply: Thanks for your kind reminder.

We have revised the statistical distribution of the I_{2D}/I_G for the three types of graphene (as-transferred, annealing and PAT) instead of the sole PAT results in the new Fig. 2e (Figure R15d). We have also supplemented the description in the figure title of Fig. 2e and corresponding main text, as follows,

Page 6, figure title of Fig. 2e: “e, Statistical distribution of I_{2D}/I_G in $100 \times 100 \mu\text{m}^2$ of the as-transferred graphene, after vacuum annealing and after PAT, inset is the corresponding Raman mapping of the graphene after PAT. The distribution of I_{2D}/I_G is homogeneous after PAT, while vacuum

annealing will cause the decrease of the values.”

Page 6 and page 7: “Subsequently, ..., are plotted in supplementary Fig. 4c-e.”

We have also supplemented the corresponding Figure R15a-b and Figure R15c as the revised supplementary Fig. 4a-b and inset of main Fig. 2e.

Figure R15. a-c, Raman mapping of I_{2D}/I_G in the region of $100 \times 100 \mu\text{m}^2$ of the as-transferred graphene, graphene after annealing and after PAT. d, Statistical distribution of I_{2D}/I_G extracted from a-c.

(7) In page 7, authors state “The QHE for the PAT graphene film is the most robust, and the values of integer plateaus are the most accurate and their quantities are also the maximum.”. The expression “and their quantities are also at the maximum” sounds imprecise. I would suggest to re-write this sentence and use another formulation such as e.g. “and plateaus are fully developed”

Reply: Thanks for your kind suggestion.

We have re-written this sentence in the revised manuscript as follows,

Page 8: “The QHE for the PAT graphene film is robust, with more emergent integer plateaus and highly accurate plateau values.”

(8) In page 8, authors state “The robust QHE plateaus under relatively mild conditions...”. Could authors explain what do they mean by “relatively mild conditions”?

Reply: Thanks for your kind reminder.

Here, the term “relatively mild” refers to the measurement conditions of magnetic field and

temperature. Compared with the results in previous papers, the measurement conditions in this study are “relatively mild”. Such as, **room temperature (300 K) QHE** of graphene on silicon wafer is first reported by A. K. Geim et al. which need the magnetic field up to **20 T** (*Science* 315, 1379-1379, 2007). In this study, we realize the RT QHE under the magnetic field of **7 T**.

For clarity, we have already re-written this sentence in the revised manuscript, as follows,

Page 8: “...**The robust QHE plateaus emerged under relatively mild measurement conditions...**”

(9) In page 10, authors state that “that means the double-layers prefer the 1 – 2 nm periodicity of Moiré patterns, as shown in Fig. 3f.” However, authors do not directly plot the periodicity of the resulting Moiré patterns in Fig. 3f but the twist angle between the two graphene layers. Although both parameters are univocally related, (for consistency/clarity reasons) I think authors should select one of the two for both main text and figure.

Reply: Thanks for your kind reminder.

The periodicity of Moiré pattern (λ) corresponds to the specific twist angle (θ) if the lattice length of two layers (a, b) are given.

$$\theta = \arccos \left[1 - \frac{a^2 b^2 - \lambda^2 (a - b)^2}{2ab\lambda^2} \right]$$

In this section, we do not further discuss the periodicity of Moiré pattern. Therefore, for clarity, we have deleted the description about the periodicity of Moiré patterns in the revised edition as follows,

Page 10: “... **We further count hundreds of double-layer graphene films which are stacked randomly and find their twist angles approximately follow the normal distribution, as shown in Fig. 3f.**”

REVIEWER COMMENTS

Reviewer #1 (Remarks to the Author):

The authors have addressed all my concerns. I support its publication on Nature Communications.

Reviewer #2 (Remarks to the Author):

The authors have mostly addressed my previous concerns. However, I have identified one remaining issue that should be addressed before I can recommend this manuscript for publication. It is necessary to provide further evidence to demonstrate whether effective interlayer coupling with atomically clean interfaces can be achieved using their methods. For instance, in Figure R5, the authors have successfully created double-layer graphene with a twist angle of near 0° ($< 1^\circ$). If the interlayer is indeed clean, the majority regions of the bilayers should spontaneously transform into Bernal stacked configurations. This transformation is characterized by distinct intensity ratios between spots in the diffraction patterns (Nano Lett. 12, 4635–4641 (2012)). I wonder if the authors have observed such a change. Although different Landau levels are presented as evidence of effective interlayer coupling, it is important to consider that band renormalization can also occur due to other factors, such as impurities. Therefore, I suggest that the authors perform optical UV-Vis spectroscopy (Phys. Rev. B 87, 205404 (2013)) to investigate if twist angle-dependent interlayer absorptions occur in the film. Such observations would provide direct evidence of interlayer coupling.

Reviewer #3 (Remarks to the Author):

I have read the authors' rebuttal letter to each of the points and the new version of the manuscript.

All the points I raised in my former report have been satisfactorily amended and/or sufficiently clarified. In particular, authors have described further the procedure used to accurately control the twist angle when stacking 2D materials at a wafer scale, which is one of the main novelties of the study.

I recommend the article for publication as it is.

Referee #1 (Remarks to the Author):

The authors have addressed all my concerns. I support its publication on Nature Communications.

Reply: We thank the Referee #1 for his/her positive comments, strong support and favourable recommendation for our work.

Referee #2 (Remarks to the Author):

The authors have mostly addressed my previous concerns. However, I have identified one remaining issue that should be addressed before I can recommend this manuscript for publication.

Reply: We thank the Referee #2 for his/her positive comments, strong support and favourable recommendation for our work.

(1) It is necessary to provide further evidence to demonstrate whether effective interlayer coupling with atomically clean interfaces can be achieved using their methods. For instance, in Figure R5, the authors have successfully created double-layer graphene with a twist angle of near 0° ($< 1^\circ$). If the interlayer is indeed clean, the majority regions of the bilayers should spontaneously transform into Bernal stacked configurations. This transformation is characterized by distinct intensity ratios between spots in the diffraction patterns (Nano Lett. 12, 4635-4641 (2012)). I wonder if the authors have observed such a change. Although different Landau levels are presented as evidence of effective interlayer coupling, it is important to consider that band renormalization can also occur due to other factors, such as impurities. Therefore, I suggest that the authors perform optical UV-Vis spectroscopy (Phys. Rev. B 87, 205404 (2013)) to investigate if twist angle-dependent interlayer absorptions occur in the film. Such observations would provide direct evidence of interlayer coupling.

Reply: Thank for your constructive and professional comment.

The effective interlayer coupling normally refers to the uniform interlayer with minimum distance for a certain twist angle. The effective interlayer coupling can be destroyed by the nonuniform interlayer distance, which is usually caused by the intercalation. **Thermal annealing at high temperature** can lead to the effective interlayer coupling, and the **stable twist angle** can be also obtained at higher temperature. Mostly carbon materials after annealing above 1800°C will show the graphitic structure with Bernal stacking. Although most twist angles are not stable and they will be changed after annealing above 400°C , like the magic-angle graphene double-layer broken after the thermal annealing (Pablo J. et al., *Nature* 556, 43-50, 2018), they still show effective interlayer coupling while their interlayer distances are uniform and minimum.

The effective interlayer coupling is usually characterized by the electrical transport, Raman spectroscopy, scanning tunnelling spectroscopy, scanning tunnelling microscope and scanning transmission electron microscope for Moiré patterns, *etc.*.

In this study, we mainly focus on the stacking transfer of a high-quality few-layer graphene superlattices, which have the effective and stable interlayer coupling for a controlled twist angle. The instable twist angles for the as-transferred graphene films are not the final products for our transfer method, because the PAT process will change the instable twist angle.

As you concerned, we have supplemented the *ex situ* Raman for the transferred double-layer graphene **annealed at different temperature**, because Raman is sensitive for the coupling change of the double-layer graphene. The double-layer graphene films are transferred on SiO₂/Si and Cu(111), and their Raman spectra after annealing at different temperature are shown in Figure R1a and R1b, respectively. The peak shape and position both become unchanged after the vacuum annealing at 450 °C, meaning that their coupling state after annealing above 450 °C is stable.

Therefore, **the PAT process in this study is performed above 450 °C, which can make sure the stable and effective interlayer coupling.** Figure R1c shows the transferred double-layer graphene film after PAT, and the Raman spectra are unchanged after the variable temperature measurements.

Figure R1. *ex situ* Raman spectra of double-layer graphene after annealing at different temperature. **a**, As-transferred double-layer graphene on SiO₂/Si. **b**, As-transferred double-layer graphene on Cu(111). **c**, PAT double-layer graphene on SiO₂/Si.

According to the published results (*Nano Lett.* 12, 4635-4641, 2012; *Nat. Mater.* 21, 1263-1268, 2022), the intensity ratio of the second-order and the first-order SEAD spots (I_2/I_1) is more than 2 only for the Bernal stacked double-layer graphene. Therefore, we have measured the I_2/I_1 ratios of the triple-

layer graphene ($\sim 20^\circ$ and $\sim 0^\circ$) for the initial Figure R5 (also as Figure R2 here). The I_2/I_1 ratios of the PAT double-layer graphene at different locations are all larger than 2, indicating that double-layer graphene with a twist angle of $\sim 0^\circ$ have the stable Bernal stacked structure.

Furthermore, it is difficult to observe *in situ* or *ex situ* SEAD pattern changes by the PAT process due to the different external environments of **suspending double-layer graphene on TEM grid**. In this study, we normally characterize the few-layer graphene superlattices with the stable interlayer decoupling after the PAT process.

Figure R2. Typical SAED patterns of the triple-layer graphene collected at random locations, intensity ratios of the second-order and the first-order SAED spots are all larger than 2, indicating the Bernal stacked structure.

As you mentioned, we have also performed the UV-Vis measurements for the as-transferred double-graphene films on the double-polished sapphire and the films after standard PAT process, shown in in Figure R3. Except for the signal noise, we do not find obvious difference for the transmittance between the as-transferred and PAT samples. All their transmittances are $\sim 95.5\%$ at 550 nm. The absorption approximately equal to the value of 100% minus the transmittance, meaning the absorption of 4.5% at 550 nm. The common UV-Vis may not be a convenient tool for experimentally observing the changes of interlayer coupling, like the electrical transport or Raman spectra.

Figure R3. UV-Vis spectra of double-layer graphene transferred onto an Al₂O₃ substrate. The as-transferred film is not performed by PAT or thermal annealing, and other two PAT samples are as the comparisons.

To better correlate the interlayer coupling with the PAT process, we have already added the intensity ratios of the first-order and second-order SAED spots from Figure R2b to the revised Figure 4e. We have also added Figure R1 as the new supplementary Fig. 12f.

We have also supplemented the following description in Page 12, Paragraph 2 as follows,

“For twist angles of roughly 0°, the intensity ratios between the second-order and the first-order SAED spots are always larger than 2 (insets of Figure 4f), indicating the Bernal stacking order^{20,49}. The variable temperature measurements confirm that the thermal annealing above 450 °C or PAT process will bring in the effective interlayer coupling and the stable twist angle (supplementary Fig. 12f).”

Referee #3 (Remarks to the Author):

I have read the authors' rebuttal letter to each of the points and the new version of the manuscript. All the points I raised in my former report have been satisfactorily amended and/or sufficiently clarified. In particular, authors have described further the procedure used to accurately control the twist angle when stacking 2D materials at a wafer scale, which is one of the main novelties of the study.

I recommend the article for publication as it is.

Reply: We thank the Referee #3 for his/her positive comments, strong support and favourable recommendation for our work.

REVIEWERS' COMMENTS

Reviewer #2 (Remarks to the Author):

The authors have addressed my concern regarding the effective interlayer interactions between the stacked graphene by providing direct evidence through TEM and optical spectra data. Now I recommend the publication of the work in Nature Communications.